# Sox9 regulates alternative splicing and pancreatic beta cell function

Sapna Puri[1,3], Hasna Maachi[1,4,5,6,7], Gopika Nair[1,8], Holger A. Russ[1,9,10], Richard Chen[1], Pamela Pulimeno[1], Zachary Cutts[2], Vasilis Ntranos [1] & Matthias Hebrok [1,4,5,6,7] ✉

Despite significant research, mechanisms underlying the failure of islet beta cells that result in type 2 diabetes (T2D) are still under investigation. Here, we report that Sox9, a transcriptional regulator of pancreas development, also functions in mature beta cells. Our results show that Sox9-depleted rodent beta cells have defective insulin secretion, and aging animals develop glucose intolerance, mimicking the progressive degeneration observed in T2D. Using genome editing in human stem cells, we show that beta cells lacking SOX9 have stunted first-phase insulin secretion. In human and rodent cells, loss of Sox9 disrupts alternative splicing and triggers accumulation of non-functional isoforms of genes with key roles in beta cell function. Sox9 depletion reduces expression of protein-coding splice variants of the serine-rich splicing factor arginine SRSF5, a major splicing enhancer that regulates alternative splicing. Our data highlight the role of SOX9 as a regulator of alternative splicing in mature beta cell function.

Pancreatic beta cell dedifferentiation is recognized as one potential cause of beta cell failure that contributes to the development of diabetes[1–7]. Dysregulation of regulatory factors and signaling pathways under conditions of stress can lead to a suboptimal beta cell, i.e., a beta cell that secretes lower amounts of insulin due to diminished activity of positive regulators, or through stabilization of negative regulators. One such factor identified by us in a rodent model of beta cell dedifferentiation was Sox9[2]. Sox9, or Sex-determining Region Y (SRY) Box 9, belongs to a family of transcriptional regulators containing a highly conserved high mobility group (HMG) domain, and is implicated in playing diverse functions in numerous organs, including maintenance of stem cell niches, chondrocyte development, cell proliferation, and extracellular matrix modeling[8,9]. Within the rodent pancreas, and in keeping with a pro-proliferative role, Sox9 promotes expansion of the

progenitor pool during embryonic development[10,11]. Loss of Sox9 early in development leads to reduction in the endocrine compartment in a dose-dependent manner[10,11]. In humans, haplo-insufficiency of *SOX9* results in campomelic dysplasia (CD), a human syndrome characterized by defective chondrogenesis and sex reversal. In patients with CD, defective pancreas morphology and dysmorphic islets suggests a similar role for SOX9 in the maintenance of the pancreas progenitor pool[12]. Although it is clear that loss of Sox9 negatively impacts endocrine differentiation, it remains unknown if Sox9 plays a role in the mature beta cell. Expressed in progenitors before the induction of the pro-endocrine regulator Ngn3[13], Sox9 is thought to be repressed through a Ngn3-dependent mechanism for the continuation of cell differentiation towards the endocrine lineage[14]. Seminal work by Gu and colleagues on Ngn3 demonstrated that factors important during

[1]Diabetes Center, Department of Medicine, University of California, San Francisco, CA, USA. [2]Graduate Program in Bioinformatics, University of California, San Francisco, CA, USA. [3]Present address: Minutia Inc., Oakland, CA, USA. [4]Present address: Center for Organoid Systems, Klinikum Rechts der Isar (MRI) and Technical University Munich, 85748 Garching, Germany. [5]Present address: Institute for Diabetes Organoid Technology, Helmholtz Munich, Helmholtz Diabetes Center, Ingolstädter Landstraße 1, 85764 Neuherberg, Germany. [6]Present address: Munich Institute of Biomedical Engineering (MIBE), Technical University Munich, Munich, Germany. [7]Present address: German Center for Diabetes Research (DZD), Ingolstaedter Landstrasse 1, 85764 Neuherberg, Germany. [8]Present address: Eli Lilly, Indianapolis, IN, USA. [9]Present address: Diabetes Institute, University of Florida, Gainesville, FL, USA. [10]Present address: Department of Pharmacology and Therapeutics, University of Florida, Gainesville, FL, USA. ✉e-mail: Matthias.Hebrok@ucsf.edu; matthias.hebrok@helmholtz-munich.de

embryonic patterning of endocrine cells but thought to be subsequently silenced in fact continue to be active in adult stages[15]. Using knock-in reporter lines, the study showed that depletion of Ngn3 in adult beta cells compromises cellular function[15], demonstrating that proteins can exert potent effects even when expressed at low levels. Our data indicate that Sox9 is one such factor that continues to be expressed in the adult beta cell and whose function sustains critical beta cell activities.

Ectopic overexpression of Sox9 in beta cells leads to cellular dedifferentiation and diabetes, with stunted expression of genes required for appropriate insulin secretion[2]. These data point to an inhibitory role of high Sox9 in beta cells. Thus, in both rodent and human, the dose of SOX9 appears critical for appropriate function. Of note, such bimodal activities are not uncommon, and similar roles have been ascribed to other pancreatic regulators, with distinct effects at discrete developmental stages. Loss of Pdx1, a crucial pancreatic regulator, leads to reduced endocrine and beta cell populations depending on the developmental stage of depletion[16,17]. In contrast, ectopic expression of Pdx1 causes erroneous reprogramming of endocrine lineages[18]. Similarly, although crucial for hormone expressing lineages[13,19,20], overexpression of Ngn3 upsets the normal distribution of endocrine subtypes within the islet[21], further confirming that levels of such regulators need precise control for appropriate organ patterning.

We demonstrate that SOX9 has a conserved, non-canonical role in pancreatic beta cells. In both rodent and human beta cells, loss of SOX9 disrupts alternative splicing, resulting in changes that negatively impact the capacity of cells to appropriately secrete insulin; thus, Sox9 is both active in adult beta cells and also plays a role distinct from its canonical DNA binding function as a transcription factor. Our data pave the way to a better understanding of the role played by Sox9 in beta cell biology and disease.

## Results

### Loss of Sox9 compromises beta cell function

The role of Sox9 during the early stages of beta cell specification is undeniable, however, its role in the mature beta cell remains unknown. We used Cre-mediated recombination to delete exons 2 and 3 from the murine Sox9 gene in the developing beta cell after endocrine specification, eliminating Sox9 expression in insulin-positive cells (*Ins-Cre;Sox9fl/fl*, referred to as *Ins-Cre;Sox9-/-*) during embryogenesis[22,23]. Beta cell-specific Sox9 knockout animals were born at expected Mendelian frequencies. The mTmG reporter mouse[24] revealed broad activity of the Cre recombinase, with Cre-positive cells expressing the green fluorescent protein that co-localized significantly with insulin (Fig. S1A, green and blue staining), along with loss of the Tomato fluorescence (Fig. S1A, red). Sox9 knockout animals exhibited glucose intolerance as they aged, mimicking the progressive loss of glucose homeostasis during diabetes (Fig. 1A and Fig. S1B). Glucose intolerant animals had elevated fasting blood glucose and increased body weight, also reminiscent of pre-diabetic phenotypes (Fig. 1A). In support of previous evidence that demonstrates a dosage effect of Sox9[10], heterozygous animals with one copy of Sox9 deleted showed an intermediate phenotype (Fig. 1A and Fig. S1B).

Prior studies have failed to detect expression of Sox9 in beta cells[2]. We have used several antibodies to test for the presence of the protein in the nuclear sub-compartment of islet beta cells via histological methods, with variable results that could be attributed to either quality of antibody or the levels of the protein that may be too low to be detected reliably in this cell type. To gain a deeper understanding of the expression of this protein, we used RNAScope technology to detect *Sox9* transcripts within the islet. As shown in Fig. 1B, *Sox9* can be found in the nuclei of insulin-positive cells in control islets, and is lost in the knockout tissue, demonstrating that Sox9 is expressed, albeit at low levels, within the adult insulin-expressing compartment. Clear accumulation of *Sox9* is visible in the ductal structures in the knockout

tissue (Fig. 1B, inset in right panel), further underscoring the specificity of the depletion of Sox9 only in the endocrine compartment.

EchoMRI analysis revealed that transgenic animals had increased fat mass and reduced lean mass as compared to age-matched control littermates, hinting at increased adipogenesis and fat accumulation (Fig. 1C). Insulin is a known adipogenic factor[25], and elevated insulin, likely the cause for the observed fat mass increase, was detected in the blood of transgenic animals (Fig. 1D). This indicates a primary beta cell phenotype that prompts a secondary peripheral fat phenotype. However, peripheral insulin resistance was ruled out as an insulin tolerance test did not reveal a difference between either cohort (Fig. S1C). Upon testing the insulin secretory capacity of control and knockout cohorts in vivo, we found that Sox9 knockout animals demonstrated increased insulin secretion at basal glucose levels, mounting a stunted secretory response (Fig. 1E, F). These data indicate that the primary defect in the knockout animals lies within the beta cells.

Beta cell dedifferentiation, defined as a reduction in critical beta cell effector genes, causes dysglycemia in animal models and impairs hormone secretory capacity in immortalized beta cells[2]. We isolated islets from control and knockout cohorts, and quantified gene expression of canonical beta cell markers, including *Ins* genes, and transcriptional regulators *Pdx1*, *Nkx6.1*, *Mafa*, and *Neurod1*. Marked reduction of Sox9 was detected in the knockouts, however, beta cell genes including *Ucn3*, a marker for maturation within beta cells, and the glucose transporter *Glut2*, which allows glucose import into the cytoplasm, were not reduced (Fig. S1D). Thus, Sox9 elimination does not interfere with beta cell identity in vivo.

In case removal of Sox9 in embryonic stages impairs beta cell growth, we directly tested for Sox9 requirement in adult beta cells. We utilized a tamoxifen (TAM)-inducible Cre that deletes *Sox9* upon drug administration (*MIP-CreERT;Sox9-/-*)[26,27]. 8-week-old animals were administered TAM and challenged with glucose after an additional eight weeks (Fig. S1E). Control animals lacked Cre for this experiment. Transgenic animals (*MIP-CreERT;Sox9-/-*) displayed a trend towards glucose intolerance, with the 60 min time point reaching statistical significance (Fig. S1F, the respective area under the curve confirms the trend towards glucose intolerance in the knockout cohort), and isolated islets showed a stunted response to high glucose as evidenced by a smaller stimulation index (Fig. S1G, H). We did not observe deregulation of basal insulin secretion in this model, suggesting perhaps that the loss of basal insulin release was a more delayed phenomenon than the 8-week time point we studied. Importantly, total insulin content in islets from both groups was unchanged (Fig. S1I), underscoring the observation that the defect in glucose clearance stemmed from the loss of hormone secretory capacity and not insulin production. As expected, quantitative gene expression analyses revealed a significant reduction in *Sox9* in islets from knockouts (Fig. S1J). A mild reduction was observed in *Ins2*, while *Ins1* remained unchanged. *Ucn3* also appeared dysregulated in the knockout cohort. Other genes, including *Pdx1*, *Nkx6.1*, *Mafa*, and *Neurod1*, did not show a significant change, hinting to retention of global cellular identity as seen in the previous model (*Ins-Cre;Sox9-/-*), and instead, compromised cellular function as the cause of the observed physiological defects. Together, these data point to a requirement for continued Sox9 function in adult beta cells.

In an in vitro approach to query the contribution of Sox9 towards beta cell function, we exposed islets isolated from adult *Sox9fl/fl* transgenic mice to adenoviruses encoding either mCherry (control) or Cre protein in culture (Fig. 1G). Six days after infection, control islets showed clear accumulation of red fluorescence in islets (Fig. 1H). Cre activity was validated using *Sox9fl/fl* islets that had the mTmG reporter. Green fluorescence served as readout for successful Cre excision (Fig. 1H, inset). A close to 50% reduction in Sox9 expression was observed in the primary islets; we only detected marginal reductions in the beta cell genes including *Ins2*, *Pdx1*, *Nkx6.1*, *Neurod1*, and *Glut2* (Fig. 1I). These data indicate that similar to what we observed in the

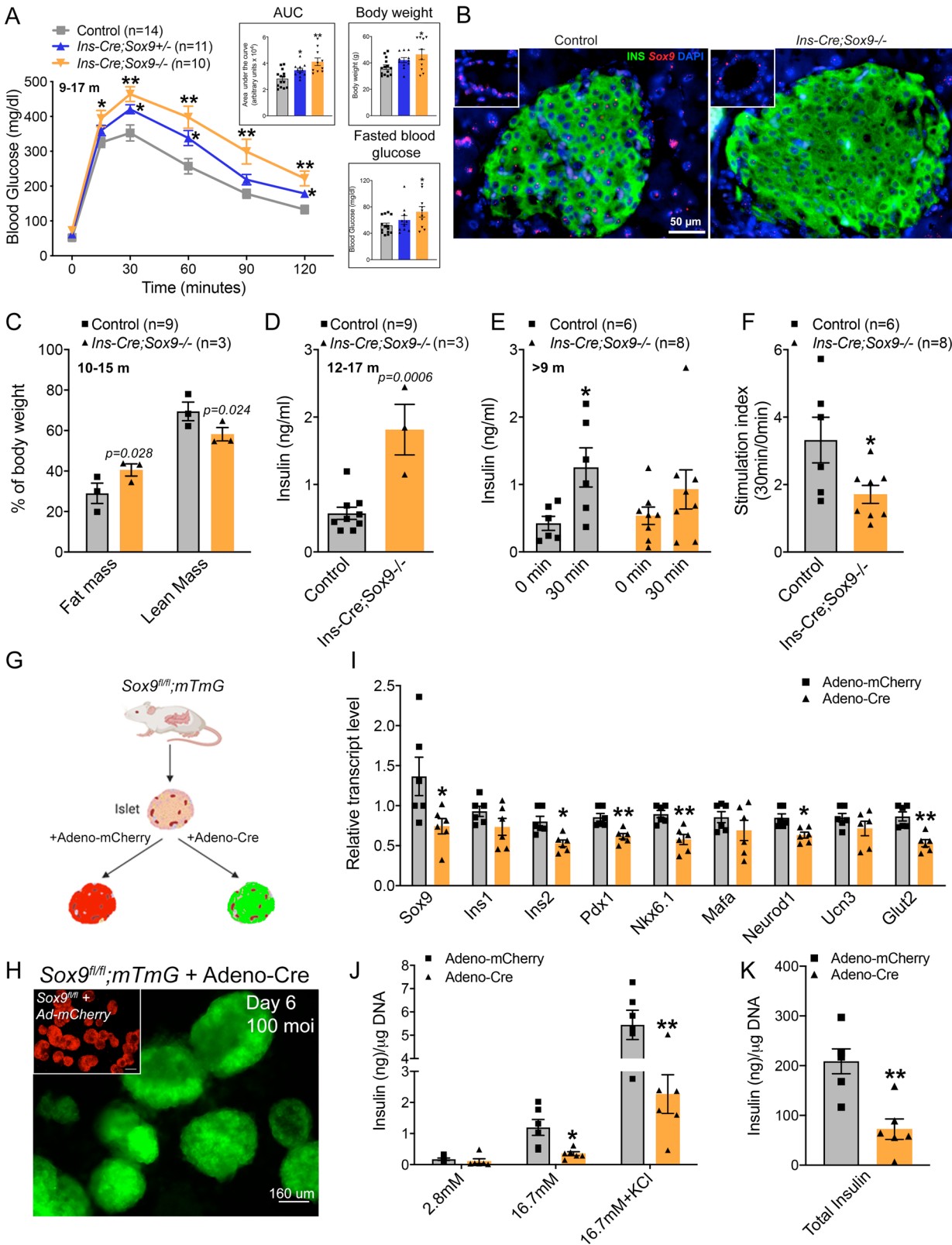

mouse model of Sox9 elimination, in vitro reduction of Sox9 only moderately deregulates beta cell genes. Importantly, when we tested the function of these islets, we uncovered a defect in hormone secretion upon Cre-mediated Sox9 elimination similar to what was found in the in vivo model of Sox9 deletion (Fig. 1J & K). Thus, studies performed both in vivo and in vitro reveal a requirement for Sox9 in adult beta cell function.

## Signaling pathways downstream of Sox9 in the beta cell

Cytological analyses of mouse knockout tissue revealed a disorganized islet structure (Fig. S2A), with increased infiltration of non-epithelial cells (marked by white arrowheads) along with perturbed packing of the endocrine cells. Immunostaining for insulin confirmed disrupted islet morphology, with loss of the compact organization of beta cells within the islet core; insulin staining was perturbed in the knockout

**Fig. 1 | Deletion of Sox9 affects beta cell function. A** Glucose tolerance test in Sox9 knockout (red, *n* = 10), Sox9 heterozygous (blue, *n* = 11), and Sox9 wild type (grey, *n* = 14) animals at 9-17 months of age shown and mean +/- SEM. The area under the curve (AUC), body weight, and fasted glucose levels are shown as mean +/- SEM. **B** RNAScope analysis of control or knockout animals (>10 months old) for *Sox9* counterstained with Insulin and DAPI was carried out three independent times. *n* = 3 controls and 3 knockout pancreata. **C** Fat and lean mass quantification for control (*n* = 9) and knockout (*n* = 3) animals at 10-15 months of age. Data shown as mean +/- SEM. P values are indicated. **D** Circulating insulin in control (*n* = 9) and knockout (*n* = 3) animals at 12-17 months of age. Data shown as mean +/- SEM. **E** A glucose challenge in vivo to measure insulin secretion in control (*n* = 6) and knockout (*n* = 8) animals at >9 months of age. Stimulation index is shown in (**F**). **G** Strategy of infecting isolated *Sox9$^{fl/fl}$* or *Sox9$^{fl/fl}$;mTmG* islets with adenoviruses.

Control samples were infected with Adeno-mCherry, while test samples were infected with Adeno-Cre. **H** Islets from adult *Sox9$^{fl/fl}$* or *Sox9$^{fl/fl}$;mTmG* animals were infected with adenoviruses encoding Adeno-Cre or Adeno-mCherry (inset) and imaged 6 days after infection. The infections were carried out in two independent experiments. **I** *Sox9$^{fl/fl}$* or *Sox9$^{fl/fl}$;mTmG* islets infected with adenoviruses encoding either mCherry (control, *n* = 6) or Cre recombinase (*n* = 6) for 6 days were processed for gene expression. **J** Glucose-stimulated insulin secretion in islets infected with either Adeno-mCherry (*n* = 6) or Adeno-Cre (*n* = 6). **K** Total insulin quantification from isolated islets infected with either Adeno-mCherry (*n* = 6) or Adeno-Cre (*n* = 6). Error bars represent S.E.M. Two-tailed Student's *t* test was used for all statistical analyses. *\**p* < 0.05, \*\**p* < 0.005. Source data are provided as a Source Data file.

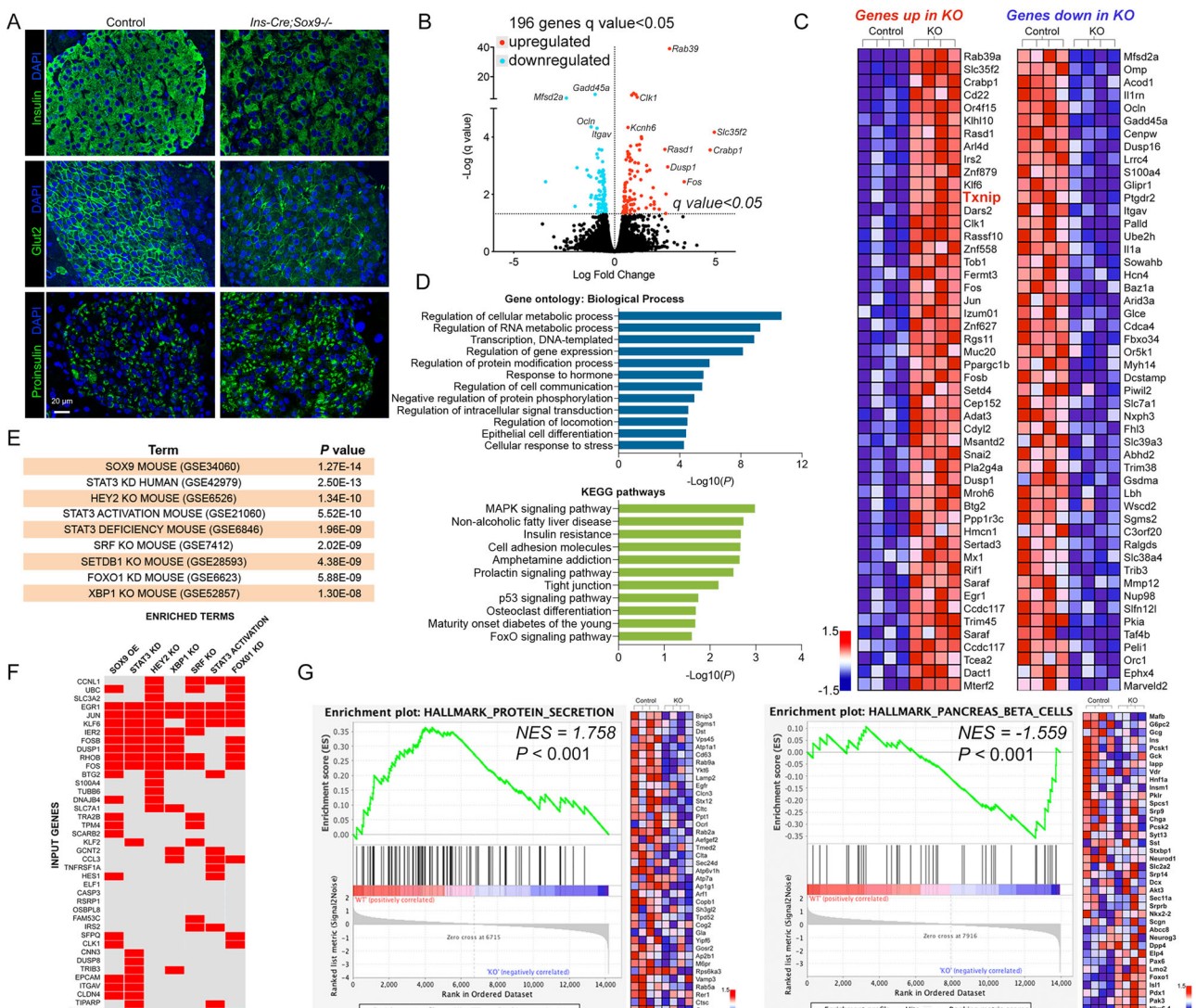

**Fig. 2 | Loss of Sox9 compromises beta cell integrity. A** Immunostaining analysis of control or *Ins-Cre;Sox9-/-* tissue from animals between 9-12 months of age. Images are representative of at least three animals per group from three separate cohorts. Insulin, Glut2, and Proinsulin staining is shown. Size bar, 20 μm. **B** RNA-seq analysis of islets isolated from control or *Ins-Cre;Sox9-/-* animals (13-16 months old, *n* = 4 per group) revealed a significant change in 196 genes. **C** Heat map of upregulated (red) and downregulated (blue) genes from the RNA-seq differential gene expression analysis is shown. Expression values are represented as colors and range

from red (high expression), pink (moderate), light blue (low) to dark blue (lowest expression). **D** Gene ontology and KEGG pathway analysis of the genes that are significantly changed in the *Ins-Cre;Sox9-/-* islets is shown. **E, F** Enrichr analysis to identify significant overlaps between genes changed in the Sox9 knockout islets and those downstream of other transcriptional regulators, including Sox9. **G** GSEA showing genes participating in the "Protein Secretion" and "Pancreas beta cell genes" gene sets. Heat maps of the genes populating these pathways are shown next to the corresponding pathway. Source data are provided as a Source Data file.

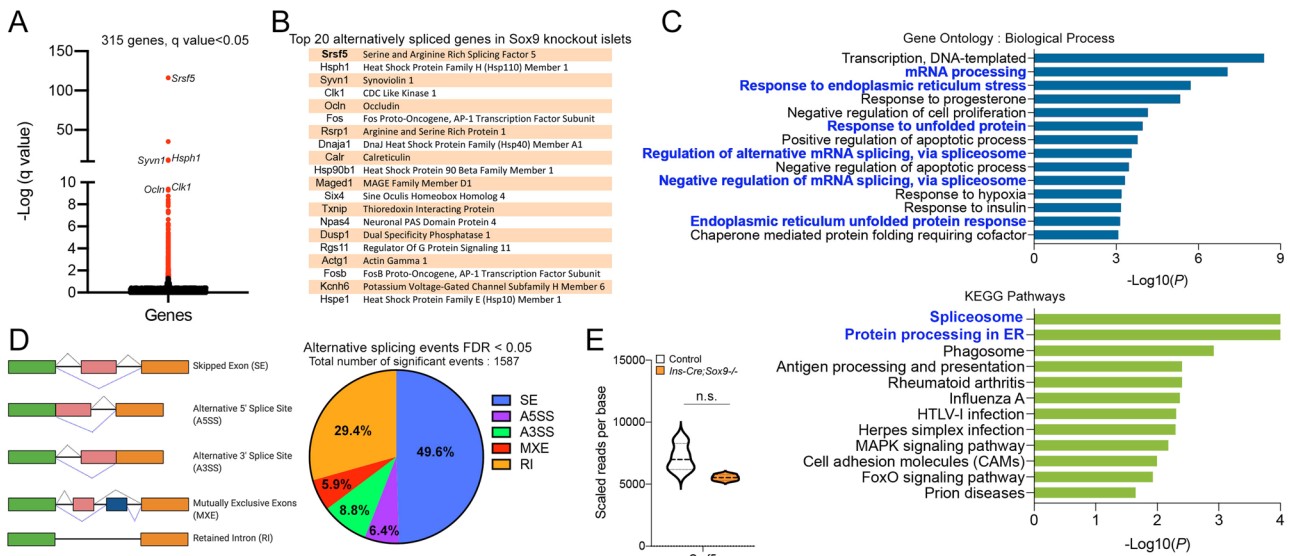

**Fig. 3 | Alternative splicing as a potential Sox9-regulated mechanism in the beta cell. A** Transcript-level alignment of mapped reads from RNA-seq data reveals genes dysregulated at the isoform level in the *Ins-Cre;Sox9-/-* islets. A volcano plot of the distribution of genes is shown. Red highlights the 315 genes with a q value < 0.05. **B** The top genes significantly alternatively spliced in Sox9-knockout samples are listed. **C** Gene ontology and KEGG pathway analysis of genes that are significantly alternatively spliced. **D** rMATs analysis of the five main alternative splicing events in the differentially expressed genes from **A**. **E** Abundance of Srsf5 at the gene level in control and *Ins-Cre;Sox9-/-* islets. Source data are provided as a Source Data file.

tissue (Fig. 2A). No overt changes were detected in Mafa, Pdx1, and Nkx6.1, while Nkx2.2 appeared to be modestly reduced (Fig. S2A). A significant reduction in Glut2 protein hinted to aberrations in translational pathways, as we could not detect a significant change in transcript levels (Fig. 2A, S1D). Finally, Proinsulin appeared elevated in islets from knockout tissue, indicating defective processing of this precursor into the mature hormone peptide (Fig. 2A). No overt differences were detected in beta cell mass when islets were stained and counted for insulin positivity (Fig. S2B). Loss of Sox9, therefore, appears to influence several functional modalities within beta cells – from glucose sensing to synthesis and processing of the insulin moiety – without inducing loss of cellular mass.

We adopted a deep sequencing approach to understand the mechanism by which Sox9 affected beta cell function (Fig. 2B). Islets isolated from adult control or *Ins-Cre;Sox9-/-* animals were processed for bulk RNA sequencing and downstream analyses using kallisto and sleuth[28,29]. A total of 16215 genes were annotated, of which 196 genes were differentially expressed (q value < 0.05) (Fig. 2B). 97 of these were upregulated, while 99 were downregulated (Supplementary Data S1). The top 50 up and downregulated genes are shown in Fig. 2C. Gene Ontology analysis of the top dysregulated genes using DAVID[30,31] revealed biological pathways "*Regulation of cellular metabolic process*", "*Regulation of gene expression*", and "*Regulation of protein modification process*" amongst others (Fig. 2D and Supplementary Data S2). Several genes deregulated in the Sox9-ko samples have key roles in pancreas and beta cell development and function (Hes1, Npas4, Rfx6, Nkx6.1, Foxo3). Gene Ontology for cellular components identified genes localized to the "Nucleus" (Hes1, Jun, Rfx6, Nkx6.1, Npas4, Txnip), "Intracellular" (S100A4, Fos, Egr1), and "Membrane-bounded organelles" (Ocln, Irs1, Cldn3, Epcam) groups (Fig. S2C, D and Supplementary Data S3). Enrichment of transcription factors and signaling molecules led to "Protein binding", "DNA binding" and "Enzyme regulator activity" as molecular functions attributed to the differentially regulated genes in the transgenic samples (Fig. S2D and Supplementary Data S4). KEGG analysis identified "MAPK signaling pathway", "Insulin resistance", "Cell adhesion molecules", "Tight junction", "Maturity onset diabetes of the young", and "FoxO signaling", among others (Fig. 2D, Supplementary Data S5, and Fig. S6). Disruption of the

MAP kinase and FoxO pathways within beta cells also have deleterious effects on the secretory machinery[32,33]. Similarly, disruption of cell adhesion and tight junction molecules compromises beta cell polarity and release of insulin granules[34]. We found dysmorphic islets in knockout tissue (Fig. S2A), and molecular confirmation of this morphological change further emphasizes the role of Sox9 as an active participant in beta cell integrity.

We used the *Enrichr* platform (https://www.ncbi.nlm.nih.gov/pmc/articles/PMC4987924/) to further compare already published data sets with those modified upon loss of Sox9 in our model, and found a significant overlap (p value of 1.2e-14) with genes expressed in the Sox9-positive population in mouse embryonic and postnatal pancreas (Fig. 2E, F and Supplementary Data S6) (GSE34060)[35]. In other words, genes known to be regulated by Sox9 from analyses of previously published datasets also populated our knockout dataset, underscoring the validity of these candidates as downstream effectors of Sox9. These genes included *Hes1, Jun, Fos, Egr1, Itgav*, and *Klf6*. Furthermore, our gene set overlapped with others derived from the manipulation of regulators of beta cell activities, suggesting interplay between these factors in various contexts (Fig. 2F and Supplementary Data S6). Elimination of Stat3, Hey2, FoxO1, or Xbp1 compromises beta cell function. Xbp1 splicing, a hallmark of how secretory protein overload is tackled within the cell, signals activation of endoplasmic reticulum-associated degradation, a pathway critically involved in insulin secretory capacity[36]. Intracellular stress that stems from protein misfolding leads to several signaling events, for e.g. targeting the high load of misfolded proteins towards degradative pathways, and an enhanced expression of chaperones that aid in proper protein folding. The thioredoxin interacting protein (Txnip) sits at a node that connects varying inputs from intracellular stress to outputs such as hormone secretion[36,37] and we find that Txnip was significantly upregulated upon Sox9 knockout (Fig. 2C). These data point to Sox9 as informing expression of multiple factors critical for beta cell functionality.

A global analysis of gene changes occurring between the knockout and control samples using the Gene Set Enrichment Analysis (GSEA) tool[38,39] identified "Hallmark Protein Secretion" at the top. The pancreatic beta cell is a secretory powerhouse, and all cellular

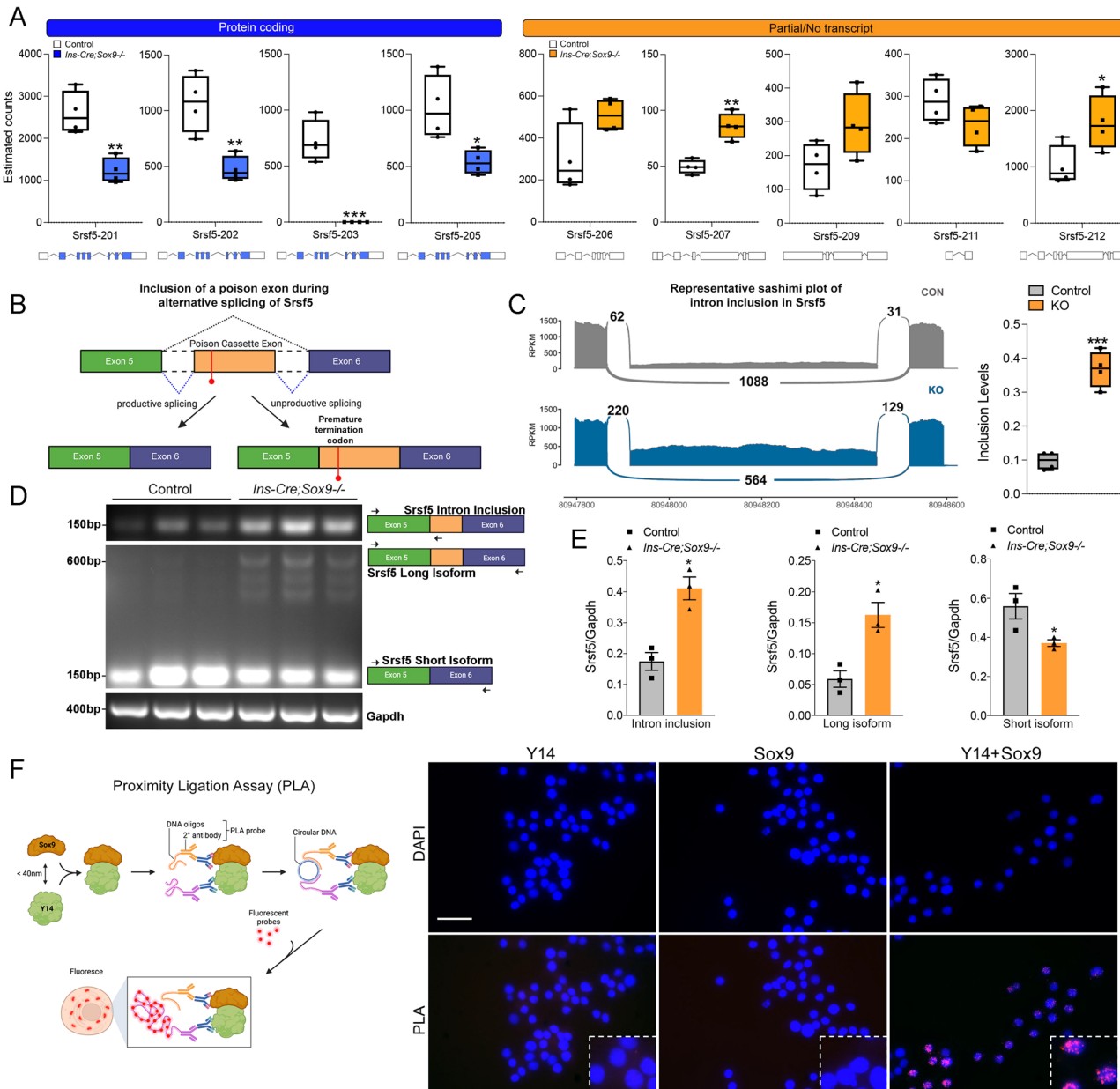

**Fig. 4 | SOX9 deletion modifies alternative splicing of Srsf5. A** Transcript level analyses of reads from the RNA-seq data showing the abundance of transcripts of coding (blue box plots) and non-coding isoforms (orange box plots) in controls and *Ins-Cre;Sox9-/-* islets. The exon compositions of the splice isoforms are shown. RNA-seq was carried out on islets isolated from control or *Ins-Cre;Sox9-/-* animals (13-16 months old, *n* = 4 per group). Data are presented as box plots (center line at the median, upper bound at 75th percentile, lower bound at 25th percentile) with whiskers at minimum and maximum values. Each dot represents one animal. **B** Splicing event in Srsf5 that leads to inclusion of a poison cassette exon with a premature termination codon. **C** Sashimi plots of alternative splicing in Srsf5. RNA-seq was carried out on islets isolated from control or *Ins-Cre;Sox9-/-* animals (13-16 months old, *n* = 4 per group). Data are presented as box plots (center line at the median, upper bound at 75th percentile, lower bound at 25th percentile) with whiskers at minimum and maximum values. Each dot represents one animal. **D** PCR validation of the comparison of abundance of transcripts with increased levels of intron 5 inclusion in Srsf5 between control and knockout islets. Primer locations for each product are shown. Controls, *n* = 3 from two independent cohorts, Knockouts, *n* = 3 from three independent cohorts. **E** Quantification of the semi-quantitative PCR shown in (**D**). RNA was isolated from control and knockout animals (*n* = 3 per group). Error bars represent S.E.M. using a two-tailed Student's t-test, *\*p* < 0.05, *\*\*p* < 0.005, *\*\*\*p* < 0.0005. **F** Proximity Ligation Assay showing a close association between Sox9 and Y14 in Ins1 cells was carried out two independent times. Size bar, 50 μm. Source data are provided as a Source Data file.

pathways converge to secrete insulin in response to elevated glucose. The enrichment plot in Fig. 2G demonstrates that genes encoding proteins involved in secretion, including Arfs, Rabs, and Vamps, are downregulated in the knockout samples. Several genes populating the "P53 pathway" were positively enriched in the knockout data set, including *Jun* and *Fos* (Fig. S2E). Genes encoding small GTP binding proteins Rab39 and Arl4d were increased, as well as Rgs11, a modifier of G-protein coupled signaling (Fig. S2F). Klf6, a transcription factor

previously implicated in beta cell function was increased in the knockout samples, as was another DNA-binding gene, *Six4* (Fig. S2F). A voltage-gated potassium channel (*Kcnh6*) and a dual specific protein kinase that phosphorylates splicing factors (*Clk1*) were also up upon Sox9 loss. Genes affecting morphological integrity, including *Itgav*, coding an integrin, and *Ocln*, encoding a tight junction protein, were downregulated upon loss of Sox9, reiterating the finding that organization of beta cells within the islets is compromised (Fig. S2F).

Peroxisomes have emerged as critical beta cell organelles and genes involved in peroxisome biogenesis were enriched in the knockout data set, implicating a role for Sox9 in peroxisomal health[40] (Fig. S2G). This global analysis of gene expression highlights the role that Sox9 plays within the beta cell; several key regulators that affect protein secretion and intracellular signaling are perturbed in the knockout tissue. However, we did not find a consistent pattern of change in genes controlling beta cell identity, including *Mafb, Ins, Neurod1, Pdx1, Isl1,* and *Foxa2* (Fig. 2G, S2H), to explain the loss of functional integrity in these islets. In summary, although no drastic changes are measured in the master regulators of insulin expression, the removal of Sox9 affects expression across several gene sets that consequently lead to compromised insulin secretion.

## Sox9 regulates alternative splicing in the beta cell

As a transcription factor, the best-known role for Sox9 is to modify gene expression by directly binding to genomic loci. Above, we described that less than 200 genes were significantly changed when Sox9 was genetically deleted in beta cells. To further investigate changes mediated by Sox9 loss in beta cells, we identified significant alterations in alternative splicing between RNA species. The kallisto-sleuth workflow increased the resolution of the analysis by highlighting changes between multiple transcripts of the same gene. 315 genes (shown in red) were significantly distinct between the knockout and control groups (Fig. 3A, Supplementary Data S7). The top 20 alternatively spliced genes in Fig. 3B have some overlap with candidates in our differential gene expression analysis, including *Clk1, Ocln, Fos, Six4,* and *Txnip*. Importantly, several other genes appeared on this list, including *Srsf5, Hsph1, Hsp90b1, Dnaja1,* and *Maged1*. These genes were not candidates for expression level differences, underscoring the information gathered by evaluating transcript-level differences. Three overarching biological processes emerged upon gene ontology analysis – *"DNA transcription", "ER stress and the unfolded protein response"*, and *"Regulation of mRNA splicing* via *the spliceosome"* (Fig. 3C and Fig. S3A, B). Several transcriptional regulators and ER stress response proteins (*Hsph1, Txnip, Hsp90b1, Dnaja1, Syvn1*) populated this list, once again implicating Sox9 as upstream of ER homeostasis.

The spliceosome is a complex assembly containing small nuclear ribonucleoproteins (snRNPs) and other proteins that carry out alternative splicing of pre-mRNA species[41]. Greater than 94% of human genes undergo alternative splicing, increasing protein diversity within the organism[42]. Modifying the function or composition of the spliceosome, therefore, has significant impacts on the ability of the complex to successfully splice precursor RNA species, with downstream effects on gene expression and disease. Sox proteins are suggested to play a role in splicing, as determined by co-localization with splicing factors and loss-of-function approaches that disrupt splicing patterns in mammalian cells[43]. Sox9 has been implicated in regulating splicing in Sertoli cells during testes morphogenesis[44], and directly binding to RNA and RNA-binding proteins in colon tumor cells[45]. However, an in vivo connection between Sox9 and splicing, and potential regulation of splicing factors in beta cells, has not been reported previously. KEGG pathway analysis revealed the "*Spliceosome*" to be the most significantly enriched pathway in the murine RNA-seq data set (Fig. 3C and Fig. S3B). We further confirmed our transcript-level analysis by using a distinct computational tool, the replicate Multivariate Analysis of Transcript Splicing (rMATS). Unlike sleuth that tests for differential transcript expression using the estimated transcript abundances within the RNA-seq data, rMATS detects alternative splicing events at the exon/intron level[46]. Using rMATS, we assessed the distribution of five major alternative splicing types from the sequencing data with a false discovery rate of less than 0.05 (Fig. 3D and Fig. S3C). 1587 events in 1237 distinct gene loci were identified as significantly differentially spliced in the knockout samples

as compared to the control, 131 of which occurred within the genes identified as differentially expressed using Sleuth (Supplementary Data S1). It is important to emphasize that the kallisto/sleuth analysis captures both gene differential expression and splicing - while rMATS only looks for splicing, providing one potential explanation for the low overlap between the two methods. Some of these genes are shown in Fig. S3D. Importantly, 14 out of the top 20 genes (shown in Fig. 3B) are also detected by rMATS (Supplementary Data S8), with an FDR value < 0.05, making them highly significant.

By far the most significantly changed set of isoforms in our dataset using sleuth was for the gene *Srsf5* (Fig. 3B), also detected as differentially spliced using rMATS (Fig. S3D). A mild downregulation at the gene level was detected in the *Srsf5* gene when comparing the knockout and control tissues (Fig. 3E). Srsf5 is a serine/arginine (SR) rich protein that plays a pivotal role in the spliceosomal complex, promoting the recruitment of the splicing machinery to splice acceptor sites[47]. Srsf5 is part of a large family of splicing factors that are highly evolutionarily conserved[48]. Like several splicing factors within the family, Srsf5 itself undergoes alternative splicing, with specific transcripts either forming the fully functional protein or being targeted towards degradation via the nonsense-mediated degradation (NMD) pathway[48]. In our data set, kallisto estimated the abundances of nine different isoforms for Srsf5 (Fig. 4A and Supplementary Data S9). In the Sox9-depleted samples, transcripts yielding complete protein products were significantly downregulated (Fig. 4A, blue bars), while incompletely processed transcripts or those targeted for degradation (Fig. 4A, orange bars) appeared enriched. One highly conserved event that occurs in all family members of the SR genes is the possible retention of a '*poison*' exon that leads to early termination of the coding sequence[48]. In Srsf5, the poison exon lies within intron 5 between exons 5 and 6 (Fig. 4B). Inclusion of the premature termination codon targets the resulting transcript towards non-sense mediated degradation (NMD)[48]. Mapped reads were enriched in this region, with inclusion levels significantly higher in the knockout sample, as shown in the sashimi plots in Fig. 4C. We validated these findings in the mouse model; using specific primers that span exon 5 and part of intron 5 (Srsf5 intron inclusion) or exon 5, intron 5 and exon 6 (long isoform), we saw the increased accumulation of a PCR-amplified product in the Sox9 knockout samples (Fig. 4D, E), demonstrating the enrichment of the alternatively spliced product with the poison exon included. In contrast, the short isoform lacking the poison exon was enriched in control samples (Fig. 4D, E). Furthermore, Srsf5 protein localization appeared significantly perturbed in the knockout tissue (Fig. S3E), suggesting that defective splicing impairs functional properties of this splicing enhancer. Srsf5 splicing was also tested in the adult beta cell Sox9 knockout model (*MIP-Cre^ERT;Sox9-/-*), and similar findings emerged. Increased inclusion of intron 5 that harbors the poison exon was detected in RNA from Sox9-deleted islets (Fig. S3F). Like Srsf5, another key regulator of splicing, Srsf6, was similarly changed, with reduction in the transcript encoding the full-length protein and a concomitant increase in transcripts that did not code for protein (Fig. S3G, H, Supplementary Data S10). Several other Srsf family members were also alternatively spliced (Fig. S3B, D), implicating loss of Sox9 in islet beta cells as a precursor to changes in alternative splicing of Srsf factors. These data point to a function of Sox9 in regulating essential alternative splicing enhancers of the Srsf family.

To further interrogate the involvement of Sox9 in alternative splicing, we looked for close association of this transcription factor with the splicing machinery. The Proximity Ligation Assay, or PLA, is an in situ assay that allows for detection of close interactions between proteins intracellularly. Proteins less than 40 nm apart generate a detectable signal visualized as enhanced fluorescence. We tested whether Sox9 was closely associated with Y14, a protein that localizes to the splicing machinery. Y14 forms part of the Exon Junction

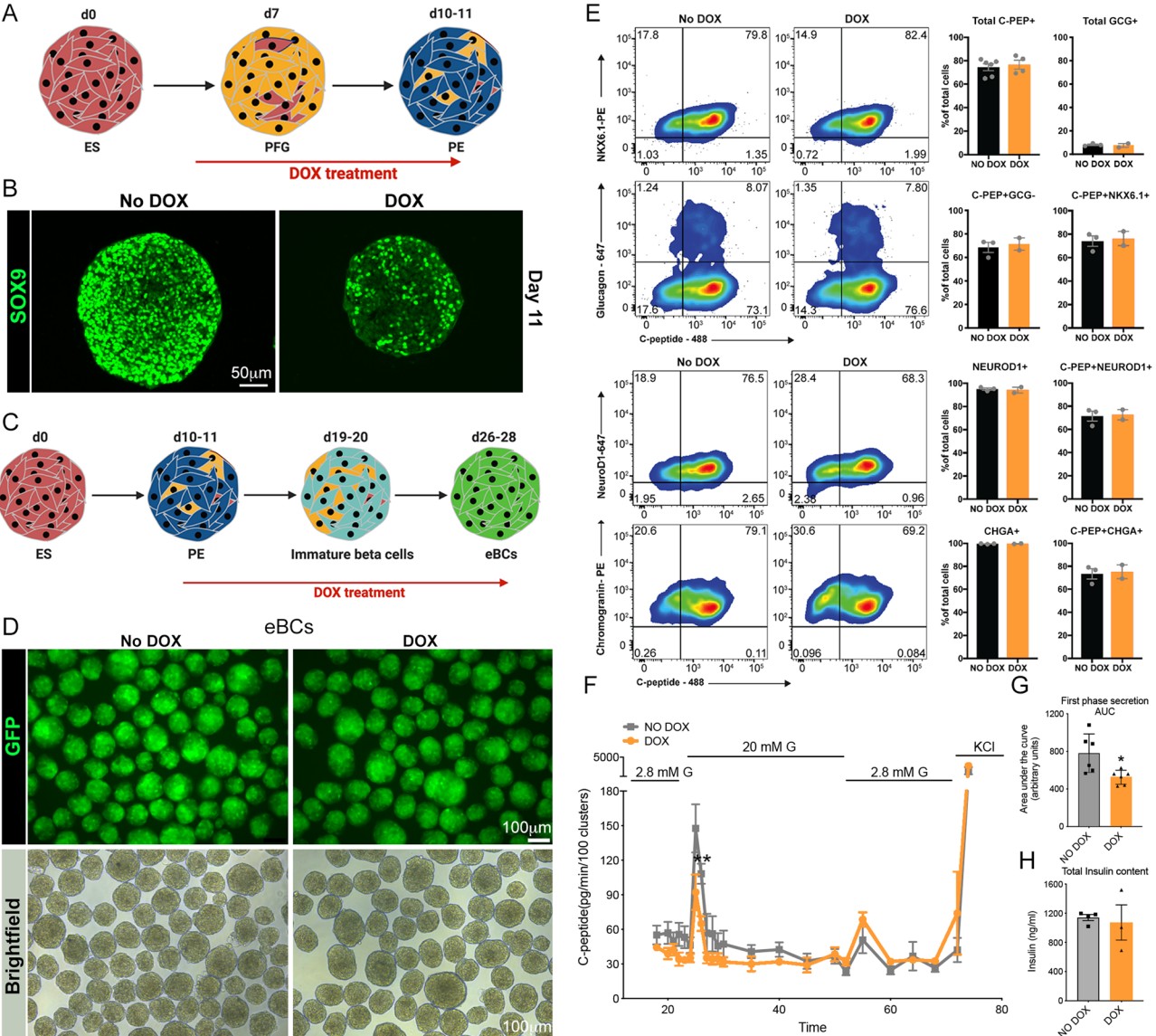

**Fig. 5 | Deletion of SOX9 in human beta cells. A** The differentiation strategy of human embryonic stem cells to pancreatic beta cells, with DOX addition. **B** Immunostaining for SOX9 at day 11 of differentiation in the absence (left panel) or presence (right panel) of DOX. The experiment was carried out at least three independent times. **C** Strategy for deleting SOX9 in endocrine progenitors. **D** eBCs generated from the iSOX9KO line in the absence (left panels) or presence (right panels) of DOX. GFP fluorescence is shown in green in the upper panel and bright field images are shown in the lower panels. The experiment was carried out at least three independent times. **E** Flow analysis on iSOX9KO cells differentiating either in the presence or absence of DOX. $n = 3$ for the NO DOX samples and $n = 2$ for the

DOX samples were used for costaining C-PEP with either NKX6.1, NEUROD1 or CHGA, and for the GCG measurements. For the total C-PEP measurements, $n = 6$ for the NO DOX samples and $n = 4$ for the DOX samples were used. **F** Dynamic Insulin secretion in control eBCs (No DOX, grey tracing, $n = 6$) and SOX9 knockout eBCs (DOX, orange tracing, $n = 6$). **G** Area under the curve calculated for the first phase of insulin secretion as shown in (**F**). **H** Total insulin content measured in control (grey bar, No DOX, $n = 4$ independent experiments) or knockout (orange bar, DOX, $n = 3$ independent experiments) eBCs. Error bars represent S.E.M. using a two-tailed Student's t test, $*p < 0.05$, $**p < 0.005$. Source data are provided as a Source Data file.

Complex (EJC) within the spliceosome[45,49]. As shown in Fig. 4F, incubation of fixed Ins1 cells with either the Y14 or Sox9 antibodies alone did not show fluorescence accumulation within the cell; however, simultaneous incubation of both antibodies showed a clear signal within the nucleus as speckles that have been shown before to be typical of splicing factors and SR proteins[50]. These data demonstrate that Sox9 is closely associated with the splicing machinery, and in combination with the functional changes in alternative splicing, confirm our conclusions that Sox9 is a regulator of splicing in pancreatic beta cells.

The secretory nature of the beta cell makes it highly susceptible to ER stress, and the deregulation of the homeostatic mechanisms that allow for insulin synthesis, folding, processing, packaging, and

secretion leads to dysfunction and disease. We found several proteins that ensure quality control of protein folding demonstrated a differential abundance of alternatively spliced transcripts upon Sox9 loss (Fig. 3B, C, Fig. S3B). *Hsph1*, *Syvn1*, *Hsp90b1*, and *Dnaja1* participate in maintaining the fidelity of protein folding and showed accumulation of non-functional transcripts and a trend towards reduction in the protein-coding isoforms (Fig. S3I-L). These data further indicate a deregulation of the intracellular pathways that maintain robust secretory capacity of beta cells as a consequence of Sox9 loss.

**Loss of SOX9 affects human beta cell function**

Little is known about SOX9 in adult human beta cells, the common belief being that this gene is absent from insulin-producing cells. To

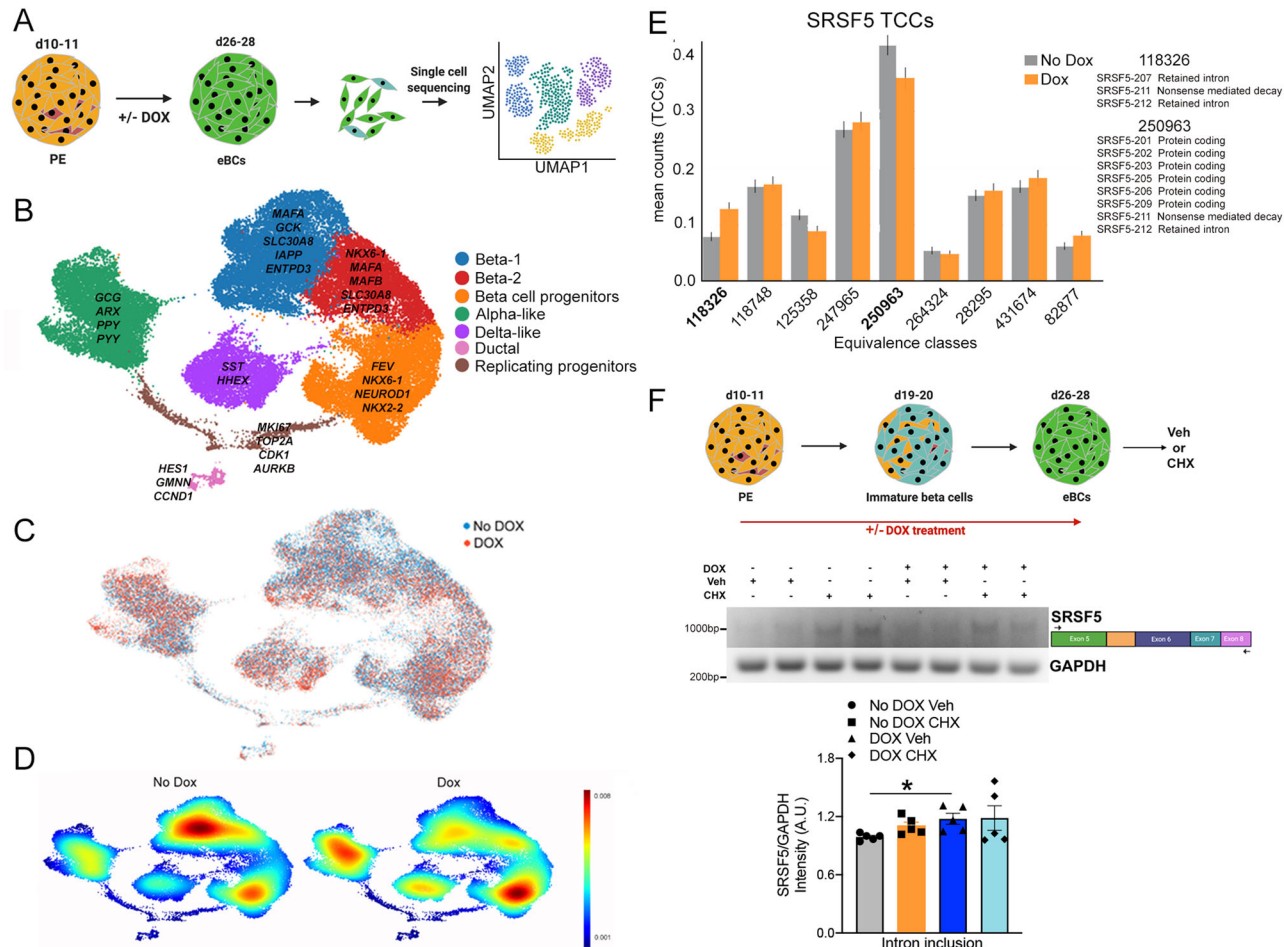

**Fig. 6 | SOX9 regulates SRSF5 splicing in human beta cells. A** Single cell sequencing of iSOX9KO cells with or without DOX from day 12 till day 28. **B** Populations identified from the single cell data. Markers used to annotate the clusters are shown. **C** Overlapping iSOX9KO eBC populations with and without DOX. **D** Fraction of cells populating different clusters with or without DOX addition. **E** SRSF5 transcript compatibility counts (TCCs) for control (grey) and DOX-treated (red) cells. Bars correspond to the mean TCCs for Dox ($n = 2392$) and for No Dox ($n = 2660$) cells in cluster 3. Error bars correspond to 95% confidence intervals around the mean estimated by bootstrapping ($n = 1000$). **F** Strategy to knockout SOX9 and inhibit nonsense-mediated degradation in iSOX9KO eBCs and test for isoform switching in *SRSF5* by PCR. No DOX Veh, grey bar; No DOX CHX, orange bar; DOX Veh, blue bar; DOX CHX, cyan bar. $n = 5$ per group. Error bars represent S.E.M. using a two-tailed Student's t test, *$p < 0.05$. Source data are provided as a Source Data file.

ascertain a role for SOX9 in human beta cells, we adopted an in vitro approach that provided reliable read outs of function. We can generate human beta cells from embryonic and reprogrammed stem cells using stepwise differentiation of pluripotent cells[51–53]. These cells are mature, functional pancreatic endocrine cells that can rescue diabetic phenotypes in vivo in a significantly accelerated time frame as compared to prior standards. Interrogating the contribution of SOX9 to mature beta cell function required the genetic deletion of this gene upon adoption of endocrine fates in our in vitro system. Generation of an inducible knockout of SOX9 in the Mel1-INS-GFP reporter line[54] is described in Fig. S4A. Using TALEN technology, cell lines constitutively expressing three guide RNAs specific to SOX9 with a doxycycline-inducible Cas9 cassette incorporated into the AAVS1 locus were generated (termed iSOX9KO). Doxycycline (DOX) enabled Cas9 transcription and successful editing within the SOX9 locus, generating an inducible knockout model (Fig. S4B, C) in three independent clonal lines examined. To determine the efficiency of SOX9 loss, we picked a developmental stage with robust SOX9 expression. SOX9 expression comes on at day 6-7 during stem cell differentiation, and is maintained at high levels till the induction of NEUROG3, at day 12 (Nair et al., manuscript in preparation). We added DOX to iSOX9KO cells from day 5 onwards, and analyzed for SOX9 on day 11 (Fig. 5A). Immunostaining

analysis on cells treated with DOX revealed a reduced accumulation of nuclear SOX9 (Fig. 5B). Next, we measured the differentiation capacity of the SOX9-depleted cells into endocrine fates. Differentiation potential of the genetically altered line showed no differences in the ability of pluripotent cells to adopt definitive endoderm and pancreatic endoderm in the absence of DOX treatment (day 0 to day 11/12). Cells were then treated with DOX from day 12 onwards (just prior to endocrine specification but after the pancreatic lineage has been determined), and cells with no DOX served as controls (Fig. 5C). Deletion of SOX9 did not affect the formation of eBCs (Fig. 5D), or the levels of C-peptide or other critical transcription factors in the clusters, as shown by the equivalent levels of GFP-positivity/marker expression in both samples (NKX6.1, NEUROD1, PAX6, ISL1; Fig. 5E, S4D–F). Thus, using this system (iSOX9KO), we can modulate the expression of SOX9 at any desired time point during human stem cell differentiation, and we see no differences in the ability of the cells to differentiate into beta cells when we remove SOX9 at the time of endocrine specification.

We next determined if the function of SOX9-KO eBCs was compromised. iSOX9KO cells were treated with DOX starting at day 12, and eBCs were generated from treated and untreated cells. eBCs were challenged with a dynamic glucose stimulation, and C-peptide level quantification in the supernatant revealed that SOX9-depleted eBCs

had a stunted first phase response to elevated glucose, as seen by the area under the curve (Fig. 5F, G). Within the time frame of the experiment, total insulin content of the eBCs with or without DOX did not change (Fig. 5H), hinting to defective sensing or secretion from these cells. In summary, our data demonstrate that SOX9 affects hormone release from beta cells in humans, a function that is conserved across species.

## SOX9 controls alternative splicing events in human beta cells

Genetic deletion of SOX9 in human cells is a powerful tool to uncover underlying mechanisms that regulate beta cell function. We adopted a single-cell sequencing approach to query how beta cells change upon loss of SOX9. eBCs treated either with or without DOX were processed for single-cell RNA-seq using the 10X Genomics platform (Fig. 6A). We found seven distinct populations within the clusters (Fig. 6B). The top 200 genes per cluster are included in Supplementary Data S11. There was a clear mature beta cell population expressing insulin and devoid of other hormones, with expression of hallmark beta cell genes *INS*, *MAFA*, *GCK*, *PDX1*, *NKX6-1*, and *NKX2-2* (Fig. 6B and Fig. S5A, C). This population can be subdivided into two populations (Beta-1 and Beta-2), both of which express *ENTPD3*, a marker for mature adult human beta cells[55]. *GCK* levels were higher in Beta-1, signaling a more mature functional state. Beta-1 also had expression of *PLIN2* and *PLIN3*, genes important for lipid droplet biogenesis[56]. A third population that shares several markers, including *NKX2-2*, *NEUROD1*, and *NKX6-1*, represents an INS + -beta cell specified progenitor-like cohort, with co-expression of *FEV*[57].

Two distinct populations showed increased levels of either *GCG* or *SST* respectively, representing the bi-hormonal populations (Alpha-like and Delta-like) that are known to be present transiently in stem cell differentiation protocols as immature cells adopt one specific identity (Fig. 6B). We also found a small cohort of ductal/progenitor cells as defined by high expression of *HES1* and *GMNN* (Geminin), as well as replication genes *CCND1* and *MKI67*. Importantly, *SOX9* was readily detected in this exocrine subpopulation, while it remained low or close to undetectable in the endocrine populations (Fig. S5D). These findings further underscore the known limitations of deep sequencing to detect genes with low-level expression. One population showed a clear decision point, with cells separating into the bi-hormonal alpha and beta cell lineage. This population of progenitors had high replicative capacity, with elevated levels of cell cycle markers such as *TOP2A* and *MKI67* (Fig. S5C). Other markers included in this population were *CDK1* (Cyclin-Dependent Kinase 1), *PBK* (PDZ Binding Kinase), *UBE2C* (Ubiquitin Conjugating Enzyme E2 C), *AURKB* (Aurora Kinase B), and *CEN* (Centromere Protein) factors, all genes involved in proliferation (Supplementary Data S11). UMAP of *SRSF5* shows that this gene is expressed ubiquitously through all populations (Fig. S5E).

Overall, all cell types were represented in both control and SOX9-depleted conditions (Fig. 6C), suggesting that during the time frame of the experiment, we do not block the specification of any one particular cell type represented in our clustering analysis. We did, however, find that treatment with DOX increased cell accumulation in the alpha and delta-like populations, as well as the beta-cell progenitors (Fig. 6D). We did not see an increase in GCG-expression cells by FACS analysis (Fig. 5E), suggesting that there is yet again a discrepancy between transcript levels and functional protein accumulation upon loss of SOX9. One possible explanation could be an accumulation of non-functional transcripts that may not translate to functional changes upon protein assessment. We re-plotted the data to represent the proportions of populations within the different clusters (shown in a side-by-side boxplot in Fig. S5F). Although the number of events is large, it is difficult to extract statistical information from such single-cell deep sequencing data due to limited biological replicates.

Differential gene expression analysis identified several genes encoding heat shock proteins (*HSPA1A*, *HSPA1B*, *DNAJB1*) that are activated by intracellular secretory stress, in keeping with our findings in the mouse (the top 200 DE genes are shown in Supplementary Data S12). Several members of the histone family (*HIST2H2BF*, *HIST1H2BG*, *HISTH4E*) were also changed, suggesting nucleosome reprograming. PTGER3, a Gi protein-coupled receptor that decreases intracellular cAMP and was shown to be elevated in type 2 diabetic islets was upregulated in treated cells[58].

We adopted a computational approach to test whether SRSF5 was differentially spliced in stem cell-derived beta cells[59]. This method relies on pseudoalignment[28] to compute Transcript Compatibility Counts (TCCs) for each cell, extracting isoform-level information from 10x Genomics data. Instead of assigning reads to genes, pseudoalignment maps reads to sets of compatible transcripts, called equivalence classes. Each equivalence class is defined by a unique set of transcripts and the TCCs represent how many reads have been assigned to it. By definition, equivalence classes are sets of reads that are multi-mapping to the same set of transcripts in a reference transcriptome[28,60]. Although there is no direct way to visualize them as a linear segment on each individual transcript, one can think of equivalence classes as the union of all subsequences of a given read length that are common between the transcripts in the multi-mapping set. To analyze our 10X scRNA-seq data, we carried out logistic regression on the SRSF5 TCCs for a pairwise differential analysis between the two groups (No DOX and DOX treatment). We saw significant isoform switching for two equivalence classes – 118326 and 250963 (Fig. 6E). Equivalence class 118326 contains reads that map to the overlapping region of isoforms encoding non-functional SRSF5 proteins (isoforms 207, 211, and 212) and was the only class that was enriched in the DOX treated samples. All three isoforms have retained introns ('poison exon') that lead to either no protein or a transcript that is targeted for nonsense-mediated degradation, thereby further strengthening our analysis of non-functional transcripts of SRSF5 being upregulated upon SOX9 deletion. These data strongly support SOX9 regulation of alternative splicing in human beta cells. Class 250963 showed the opposite isoform-switching pattern, with the transcripts populating this class showing fewer reads mapped in the presence of DOX. Further in support of what we found in the rodent model, reads populating this class mapped to several coding transcripts that generated full-length protein (Fig. 6E). These data provide evidence that even in human cells, SOX9 regulates SRSF5 splicing, and loss of SOX9 results in increased non-productive splicing of SRSF5.

To experimentally validate our bioinformatics findings, we assessed SRSF5 splicing in human stem cell-derived eBCs (Fig. 6F). Control eBCs or those treated with DOX for 16-18 days were treated with cycloheximide (CHX) to block nonsense-mediated decay and enrich for transcripts targeted for degradation. We quantified the PCR product amplified using primers spanning exon 5 and intron 5 (the poison exon included in the alternatively spliced isoforms 207, 211, and 212). As expected, treatment of controls with CHX (no DOX) showed an increased abundance of messages that contain the retained intron (Fig. 6F). Notably, reduction of SOX9 via DOX treatment alone led to a marked increase in the amplicon when compared to controls treated in the absence of DOX, indicating enhanced presence of isoforms containing the poison exon. These data show that as in mouse, loss of human SOX9 switches alternative splicing of SRSF5 from a productive to a non-productive mechanism, showing that this transcription factor regulates a conserved, critical mechanism to control gene expression in pancreatic beta cells.

## Discussion

Sox9 plays a critical role in the expansion and maintenance of progenitors during pancreas and endocrine development[10,11,61,62]. The only defined function of Sox9 in the endocrine pancreas is during embryonic development in rodents; less is known about this protein in human cells. Mutations in SOX9 lead to campomelic dysplasia (CD), a

congenital disease characterized by bowing of bones as well as other skeletal and reproductive defects[63]. SOX9 is found within the expanding pancreatic epithelium during human embryogenesis[2,12,30,31], and patients with CD show endocrine defects including islet dysmorphia and compromised expression of some markers. This defect is likely due to reduced progenitor populations in the developing epithelium, though this hypothesis remains to be tested. Using our iSOX9KO line, we can interrogate a temporal requirement of this factor in early human pancreas specification, including ductal and acinar lineages.

Sox9 has failed immunodetection in adult islet beta cells, and positive staining by in situ hybridization suggesting presence of *Sox9* mRNA has been attributed to background from surrounding cells. Anti-sera against Sox9 have been unreliable, with non-specific staining confusing the clear identification of cells with low levels of this protein. We have been unable to use available antibodies to get a consistent answer to whether Sox9 is present in human and rodent beta cells. However, we believe that this is a technical limitation, and low levels of gene expression persist in rodent and human cells that are sufficient to effect downstream mechanisms and influence cell function. Similar phenotypes have been seen when Ngn3, undetectable by immunostaining, was knocked out in adult beta cells, suggesting that low-level activity is sufficient for some proteins to exert effects.

We adopted a Cre-lox knockout approach to delete Sox9 in mouse cells at the time of beta cell specification, and found that with time, transgenic animals developed glucose intolerance, similar to what is seen in patients with T2D. Progressive erosion of functional capabilities suggests that the changes downstream of Sox9 deletion are not sledgehammer modifications such as removing Pdx1. Beta cell mass did not change between control and knockout islets, further underscoring a functional block and not cell death as the cause of the phenotype. Changing the landscape of alternative splicing, with accumulation of non-functional transcripts of critical genes, and activation of chronic stress response pathways in response to aging can all lead to compromised beta cell function. Removing SOX9 acutely in human stem cell-derived beta cells led to a blunted first phase insulin secretion, suggesting that similar phenotypes, albeit at varying amplitudes, result from removal of SOX9 in beta cells. The finding that the time frame of our experiment does not trigger drastic changes upon SOX9 removal in human beta cells is concordant with rodent data, where most changes at the gene expression level are modest. These findings indicate a conserved role of SOX9 in both rodent and human beta cells. As evidenced by our protein staining and FACS data, we did not observe any significant changes in the endocrine lineage between control and doxycycline-treated populations, similar to the rodent genetic loss model.

Significantly, we found that Sox9 appeared to regulate the splicing of important splicing regulators. Srsf5 is a member of a large family of serine/arginine rich (SR) proteins that control splicing in cells by promoting exon inclusion to expand protein diversity[64]. Several SR proteins are ultraconserved (stretches of at least 200nt that are conserved/identical between human and rodents)[65], with long stretches of sequence identity between species[66]. In humans, 11 human SR genes are alternatively spliced to retain a poison cassette exon that upon inclusion leads to an early stop codon, rendering the transcript noncoding by targeting it to the nonsense-mediated pathway for degradation. This results in the downregulation of the gene by targeting a fraction of the gene's pre-mRNA towards decay. Unproductive splicing of Srsf5 and other family members is a highly conserved event[48].

Alternative splicing expands proteomic diversity in a context-specific and temporal manner, and has recently emerged to play a significant regulatory role in the beta cell. Several reports describe the important function of alternative splicing in regulating insulin secretion[67]. Srsf6, a closely related SR protein, has been implicated in beta cell function, as loss of this factor causes decreased insulin

secretion, endoplasmic reticulum stress, and apoptosis due to mitochondrial defects[68]. In human EndoC cells, exposure to stress-inducing agents leads to a modification in the alternative splicing of SR proteins, accompanied by dysregulation of key beta cell genes[69]. Approximately 25% of all splicing events were dysregulated in islets isolated from patients with T2D, further underscoring the significance of this pathway in the maintenance of Insulin secretory function[69]. The significance of our data lies in the observation that Sox9 loss triggers splicing towards non-functional protein isoforms, as exemplified by the SR factors and ER chaperones. Such a switch from productive to loss-of-function phenotypes in important regulatory cascades points to a conserved, critical role of Sox9 in preserving functional homeostasis in the beta cells.

A remaining question is the exact mechanism by which Sox9 affects alternative splicing. Sox9 has been implicated in roles other than that of a transcriptional regulator in tissues such as testes and colon cells[43,44] and clear accumulation of Sox9 in the cytoplasm of alpha cells has also been noted[70]. In colon cancer cells, SOX9 was predicted to modify splicing patterns by its association with Y14, an RNA-binding protein that forms part of the exon junction complex[45,49]. Our data showed a similar colocalization of Sox9 with Y14 in rodent cells (Fig. 4), further pointing to an involvement of Sox9 in the splicing machinery. In addition to its canonical nuclear localization, Sox9 has been demonstrated to be associated with ribonucleoprotein transcripts in an RNA-dependent manner, colocalizing with SR proteins[71]. In Xenopus oocytes, Sox9 colocalizes with SR proteins in lateral loops of the condensed chromatin, suggesting that Sox9 participates in splicing events. It is likely that a similar role for Sox9 in maintaining productive splicing exists in these other tissues as well, including endocrine islets. However, considering the very low levels of Sox9 protein in beta cells, determining such interactions with other splicing regulators has not been achieved.

Our data show that Sox9 continues to play important roles in regulating function within the beta cell, and it does so in part by regulating an important mechanism within the cell that perturbs the secretory pathway, consequently blocking correct functioning of the cell. Further investigation should shed light on what other players downstream of Sox9 contribute towards modifying splicing in beta cells.

## Methods

### Animal strains and ages, intraperitoneal glucose tolerance test (IPGTT), insulin tolerance test, and hormone measurement in vivo

All animal studies were approved and carried out in compliance with UCSF IACUC protocols (Protocol number AN110127). Mice were maintained at 21–23 °C with 30–70% humidity and a 12 h dark/light cycle in a pathogen-free facility at UCSF. Mice were provided with food (5058, PicoLab) and water ad libitum and were housed in autoclaved individually ventilated caging in groups of up to 5 mice per cage. Mice were bred and grown in-house and maintained in the same facility including the following strains: Sox9[flox/flox] mice - Strain: C57Bl6/J, B6.129S7-Sox9tm2Crm/J; (JAX# 013106); lns-Cre - Strain: mixed (Dr. Pedro Herrera's laboratory); MIP-Cre[ER] - Strain: C57Bl6/J, B6.Cg-Tg(Ins1-cre/ERT)1Lphi/J; (JAX# 024709); and mTmG (also called ROSA[mT/mG]) Strain: Gt(ROSA)26Sortm4(ACTB-tdTomato,-EGFP) Luo/J; (JAX# 007676). For Cre-positive animals, males were used for physiological analyses. *Sox9[fl/fl]* (denoted as *Sox9-/-* throughout), *MIP-Cre[ER]* and mTmG mice have been described previously[24,26,72]. *Ins-Cre* mice were obtained from Dr. Pedro Herrera's laboratory[22]. Male mice (7 – 9) months old were used for all experiments with MIP-Cre[ERT] mice, whereas male mice (9 – 17) months old were used for all experiments with *Ins-Cre* mice, except in the case of glucose-stimulated insulin secretion in isolated islets infected with adenoviruses (Fig. 1), where a mix of male and female mice was used. We observe gender differences

in our animal colonies (for all the *Cre-Sox9* knockout cohort, with a small n number, we do not see a phenotype in the females, and have restricted our analysis to males). The ages of all animals are included: Glucose tolerance test: 9-14 months old males, Fat and lean mass quantification: 10–15 old months males, Circulating insulin levels: 12-17 months old males, In vivo glucose challenge: > 9 months old males (controls included *Sox9+/-* and *Ins-Cre;Sox9+/-* animals), Ex-vivo glucose stimulated insulin secretion: 12-22 months males and females, RT-QPCR analysis: > 12 months old males, RNAscope analysis: > 10 months old males, Immunostaining experiments: 9-12 months old males, RNAseq experiments: 13-16 months old males. For the IPGTT, after a 16-18 hour fast, mice were weighed, blood glucose levels measured using the Contour Glucometer, and injected intraperitoneally with a 1 M glucose solution at 10 µl per g body weight. Blood glucose was measured every 30 minutes for 2 hours after injection. For ITT, mice (7–9 months old) were fasted for 16-18 hours, and body weight and blood glucose were measured. Insulin was injected intraperitoneally with a dose of 1U/kg of body weight. Blood glucose was measured every 30 minutes for 2 hours after injection. For in vivo hormone measurement, blood was collected before and after a glucose challenge from the tail vein, spun down to collect serum, and stored at −80 °C with protease inhibitors (Roche). Insulin concentration was calculated using the Insulin Rodent Chemiluminescence ELISA kit (ALPCO).

### Glucose stimulated Insulin secretion from islets
10 size-matched islets (control or transgenic) were incubated in KRB with 2.8 mM glucose for 30 min with gentle shaking. The supernatant was discarded and islets were incubated in KRB with 2.8 mM for one hour. The supernatant was collected and frozen at −20 °C with protease inhibitors. Islets were subsequently incubated with 16.7 mM glucose for one hour followed by 16.7 mM glucose and 100 µM IBMX for another hour. Supernatants were collected and stored at −20 °C with protease inhibitors. Total insulin was extracted overnight in acid/alcohol buffer, followed by DNA extraction[73]. Measurement of Insulin was carried out using the Insulin Rodent Chemiluminescence ELISA kit (ALPCO).

### Histology, immunohistochemistry, and islet size quantification
For immunohistochemical analyses, pancreata were dissected from transgenic mice followed by overnight fixation at 4 °C in Z-Fix (Anatech) which was prepared as per the manufacturer's instructions. After fixation, tissues were washed to remove fixative, processed for paraffin embedding following standard procedures, and then sectioned using a microtome into 5-6 µm sections for further processing. Prior to staining, slides were de-paraffinized by incubation in Xylenes (100%) followed by rehydration in ethanol (100%, 95% and 70% sequentially). To unmask antigens, slides were boiled in a water bath in Citrate buffer for 5 min, followed by no boiling for 30 s, and another 3 min of boiling. After cooling to room temperature, slides were washed in water followed by PBS, and incubated in blocking reagent (1% BSA in PBS-tween) for 30 min at room temperature to block non-specific antigens, and then incubated with the appropriate primary antibodies in the blocking reagent overnight at 4 °C in a hydrated chamber. After primary incubation, slides were washed three times in PBS for 5 min each, followed by incubation for 30 min at room temperature in the appropriate secondary antibodies in blocking reagent. For immunofluorescence, slides were washed in PBS (three times for 5 min each) and mounted using ProLong™ Gold Antifade Mountant (Thermo Fisher Scientific) mounting medium with DAPI. For immunohistochemistry, slides were further incubated for 30 min in ABC solution (Vectorlabs), washed in PBS, and developed using DAB reagent (Vectorlabs) as per manufacturer's instructions. The primary antibodies used were: mouse anti-Insulin, 1:500 (I2018, Sigma); guinea pig anti-Insulin, 1:500 (#A0564, Dako); rabbit anti-Glut-2, 1:500 (#07-1402,

Millipore); mouse anti-proinsulin, 1:200 (clone GS9A8, Developmental Studies Hybridoma Bank, University of Iowa); rabbit anti-Pdx1, 1:200 (07-696 Millipore); mouse anti-Nkx6.1, 1:200 (F55A10-c, Developmental Studies Hybridoma Bank, University of Iowa); rabbit anti-Mafa, 1:200 (IHC-00352, Bethyl labs); mouse anti-Nkx2.2, 1:200 (74.5A5-c, Developmental Studies Hybridoma Bank, University of Iowa); chicken anti-GFP, 1:500 (GFP-1020, Aves); rabbit anti-RFP, 1:500 (600-401-379, Rockland); rabbit anti-SOX9, 1:2000 (Sigma, Prestige HPA001758). Primary antibodies were detected with Alexa-488, Alexa-555 and Alexa-633 conjugated secondary antibodies (#A11029, #A11034, #A11073, #A21428, #A21435, #A21422, #A21105, Invitrogen) or biotinylated anti-rabbit (#111-065-003, Jackson ImmunoResearch) and anti-guinea pig (#BA-7000, Vector Labs) antibodies, all used at 1:200 dilutions. For flow analyses on human stem cells, the primary antibodies used were: Human anti- PAX6-Alexa647, 1:50 (#562249, BD Bioscience); Islet-1-PE, 1:50 (#562547, BD Bioscience); NKX6.1-Alexa647, 1:50 (#563338, BD Bioscience); NKX2.2-PE, 1:50 (#564730, BD Bioscience); Chromo-graninA-PE, 1:50 (#564563, BD Bioscience); NeuriD1-Alexa647, 1:50 (#563566, BD Bioscience); Glucagon-Alexa647, 1:2000 (#G2654, Sigma, mouse antibody conjugated in-house); C-peptide-488, 1:200 (C-PEP-01, Chemicon, mouse antibody conjugated in-house); Human SOX17-Alexa488, 1:50 (#P7-969, BD Bioscience); Human FOXA2-PE, 1:50 (#N17-280, BD Bioscience). For islet mass quantification, sections 100 µm apart were stained with anti-insulin antibody, and total islet area quantified as a percent of total pancreatic area. Bright field images were acquired using a Zeiss Axio Imager D1 microscope. Zeiss Axioscope2 wide field and Zeiss ApoTome microscopes were used to visualize fluorescence. Unless otherwise noted, all photomicrographs shown are representative of at least three independent samples of the indicated genotype.

### Adenovirus infection
Islets were isolated from either *Sox9fl/fl* or *Sox9fl/fl;mTmG+/-* adult male mice. Adeno-Cre (VVC-U of Iowa-5) and Adeno-mCherry (VVC-U of Iowa-537) were both 10e7 titer. Per 50 islets, 0.5ul of virus was added in 2 ml of media (RPMI, NEAA, Penicillin and Streptomycin, HEPES) without serum. After a 2 hour incubation, 200ul of serum was added and the islets were then incubated for 6 days. A modified protocol was used for the insulin secretion assay - islets were isolated from *Sox9fl/fl* female adult mice and then partially dissociated using 1 mM EGTA treatment to ensure the penetration of the virus into the islet core. After isolation, the islets were washed 2x with 1 ml dissociation buffer (1× Hank's balanced salt solution or HBSS, 20 mmol/l HEPES, 5 mmol/l glucose and 1 mmol/l EGTA) and then incubated in 0.5 ml of the dissociation buffer for 3 min at 37 °C. Islets were then infected with Adeno-mCherry or Adeno-Cre (1 ul/100 islets) overnight, after which the medium was replaced with a complete medium and cultured for an additional 5 days. For the static incubation assay, on day 6 after the adenoviral infection, 10 size-matched infected islets (Adeno-mCherry or Adeno-Cre) were incubated twice in 1 ml KRB containing 0.1% (w/v) FA-free BSA and 2.8 mM glucose for 20 min at 37 °C. Islets were subsequently incubated for 30 min in 0.5 ml of KRB containing 2.8 mM glucose, 16.7 mM glucose, followed by 16.7 mM glucose + 40 mM KCl. The supernatant was collected and frozen at −20 °C with protease inhibitors. Total insulin was extracted overnight in acid/alcohol buffer, followed by DNA extraction[73]. Insulin was measured using the Insulin Rodent Chemiluminescence ELISA kit (cat# 80-INSMR-CH01).

### In situ hybridization using RNAscope
RNA staining was performed with an RNAscope Multiplex Fluorescent Reagent Kit v2 (cat# 323136) according to the manufacturer's instructions. Paraffin-embedded slides were dewaxed using 2 × 5 min xylene, 2 x 2min 100% EtOH and then air-dried. Antigen retrieval was performed using RNAscope 1X Target Retrieval Reagents heated up to 98 °C for 15 min. Slides were washed with double-distilled water, then

100% EtOH for 2 min and air dried. RNAscope Protease III was applied to the dry slides and incubated for 30 min at 40 °C in the oven and then rinsed with RNAscope 1X Wash Buffer. The slides were incubated with the RNAscope Probe - Mm-Sox9 (cat# 401051) for 2 h at 40 °C. The probe was amplified using RNAscope Multiplex FL v2 AMP 1 for 30 min, then RNAscope Multiplex FL v2 AMP 2 for 30 min, followed by RNAscope Multiplex FL v2 AMP 3 for 15 min, and RNAscope Multiplex FL v2 HRP-C1 for 15 min. All amplification steps were performed at 40 °C, and slides were washed between steps using RNAscope 1X Wash Buffer. Signal was developed using the TSA Plus Cyanine 3 (cat# NEL744001KT) 1:700 for 30 min at 40 °C and blocked with HRP Blocker for 15 min at 40 °C. The slides were mounted using the Pro-Long™ Glass Antifade Mountant and the images acquired using the 40X Zeiss Apotome.

## Proximity ligation assay

The association between Sox9 and Y14 was assessed using the in-situ proximity ligation assay (PLA) (DUO92101-1KT, Sigma-Aldrich). This method enables the detection and visualization of protein interactions within cells. Ins1 cells were obtained from Dr. Chris Newgard's lab at Duke University under MTA agreement, plated on glass coverslips, fixed with 4% paraformaldehyde for 15 minutes at room temperature, and then permeabilized in 0.5% Tween in PBS for 30 minutes. Cells were blocked with the blocking solution provided in the PLA kit for 1 hour at 37 °C according to the PLA protocol, followed by incubation overnight at 4 °C with primary antibodies (1:200) against SOX9 and Y14. Duolink PLA probe incubation, ligation, and amplification were then performed according to the manufacturer's recommendations. Anti-SOX9 (HPA001758, Sigma-Aldrich) and anti-Y14 (05-1511, EMD Millipore) antibodies were used at 1:200 concentration.

## Human stem cell culture and analyses

The experiments were performed using the NIH-approved human embryonic stem cell (hESC) line MEL-1 (NIH approval number: NIH hESC-11-0139) in accordance with NIH guidelines, and approved by the UCSF Committee on Human Genes, Embryos and Stem Cell Research (GESCR) (#11-05307). None of the cell lines used in this study were reported to the ICLAC registry. The human embryonic stem cell line used in this study, Mel1 INSGFP/W, was obtained from Dr. Ed Stanley at Monash University, Australia, and Dr. Stanley provided consent to use these cells under the MTA. The iSOX9KO line was generated from the parental Mel-1 INSGFP/W line. iSOX9KO cells were maintained and expanded in human embryonic stem cell (hESC) media on mouse embryonic fibroblasts (MEFs). Cells were enzymatically dissociated using TrypLE (Gibco). Cell integrity was ensured by regular checks for mycoplasma contamination. For differentiation, hESC cultures at confluence were seeded in 6-well suspension plates with 5.5 ml hESC medium supplemented with 10 ng/ml ActivinA (R&D Systems) and 10 ng/ml HeregulinB (Peprotech) at a density of $5.5 \times 10^6$ cells per well. Plates were incubated at 37 °C in 5% $CO_2$ on an orbital shaker at 100 rpm to promote 3D sphere formation. After 24 hours, spheres were collected in 50 mL Falcon tubes, gravity settled, washed with PBS, and resuspended in day 1 media. The resuspended spheres were evenly distributed in new 6-well suspension plates containing 5.5 mL of media per well. Thereafter, media was changed everyday. Till day 3 of the differentiation protocol, spheres were fed by removing 5 ml of the media and replenishing with 5.5 ml. From day 4 till day 20, the cells were fed by removing 4.5 ml of media daily and addition of 5 ml of fresh media. For differentiation of the human cell line included in this manuscript, we followed our previously developed protocol for media compositions to ensure the reproducibility of our results. The compositions are noted here: Day 1: Base media RPMI (Gibco) with 0.2% FBS, 1:5,000 ITS (Gibco), 100 ng/ml activin A, and 50 ng/ml WNT3a (R&D Systems). Day 2: Base media RPMI with 0.2% FBS, 1:2,000 ITS, and 100 ng/ml activin A; Day 3: Base media RPMI with 0.2% FBS, 1:1,000 ITS,

2.5 μM TGFbi IV (CalBioChem), and 25 ng/ml KGF (R&D Systems); Day 4-5: Base media RPMI with 0.4% FBS, 1:1,000 ITS, and 25 ng/ml KGF; Day 6-7: Base media DMEM with 25 mM glucose containing 1:100 B27 (Gibco), 3 nM TTNPB (Sigma); Day 8: Base media DMEM with 25 mM glucose containing 1:100 B27, 3 nM TTNPB, and 50 ng/ml EGF (R&D Systems); Day 9-11: Base media DMEM with 25 mM glucose containing 1:100 B27, 50 ng/ml EGF, and 50 ng/ml KGF; Day 12-20: Base media DMEM with 25 mM glucose containing 1:100 B27, 1:100 Glutamax (Gibco), 1:100 NEAA (Gibco), 10 μm ALKi II (Axxora), 500 nM LDN-193189 (Stemgent), 1 μm Xxi (Millipore), 1 μM T3 (Sigma-Aldrich), 0.5 mM Vitamin C, 1 mM N-acetyl Cysteine (Sigma-Aldrich),10 μM zinc sulfate (Sigma-Aldrich) and 10 μg/ml of Heparin sulfate. On day 20 of differentiation, spheres were dissociated with Accumax into single cells for flow cytometry. Live GFP$^{high}$ cells were sorted on Aria II at low flow rates and reaggregated in Aggrewell™-400 (StemCell Technologies) at 1000 cells/cluster in CMRL containing 1:100 B27 (or 10% FBS), 1:100 Glutamax (Gibco), 1:100 NEAA (Gibco), 10 μm ALKi II (Axxora), 0.5 mM Vitamin C, 1 μM T3 (Sigma-Aldrich), 1 mM N-acetyl Cysteine (Sigma-Aldrich), 10 μM zinc sulfate (Sigma-Aldrich) and 10 μg/ml of Heparin sulfate. On day 22-23, the reaggregated enhanced Beta-clusters (eBCs) were transferred from Aggrewells and placed on orbital shakers at 100 rpm, and further cultured for up to two weeks. Media was changed every third-day following reaggregation. To induce SOX9 deletion, 2ug/ml doxycycline was added on the indicated days and replenished fresh when the media was changed. Flow analyses were carried out to ascertain differentiation success and dynamic insulin secretion on eBCs[53]. The gating strategy is included in the Supplementary Information.

## Generation of the iSOX9KO cell line

Generation of the iSOX9KO cell line was approved by the Committee on Human Research at the University of California, San Francisco. The INS-GFP MEL-1 line served as the parental line for the iSOX9KO line, which was generated by targeting the endogenous AASV1 locus using TALEN mediated recombineering[74]. Specifically, one targeting construct contains a CAG-M2rtTA, t2A-NEO cassette, while the second cassette contains a TRE-Cas9, three U6-gRNAs, and t2A-PURO sequence. This approach enables the selection of integration of each construct into one of two alleles. The 3 guide RNAs targeting the endogenous SOX9 gene have the following sequences: #1 GCTCGCCGATGTCCACGTCG, #2 GCCGCCCAACGGCCACCCGG, #3 GTGCGCCTGTGGCTGCTGCG. After characterizing the initial three clones, all remaining experiments were carried out on clone 2. The T7 endonuclease assay was performed according to the manufacturer's instructions (NEB, E3321).

## Quantitative PCR and gene expression array

For gene expression analyses, all cells (cell lines and islets) were processed using the RNeasy kit (Qiagen) for cell lysis followed by column-based RNA isolation as per the manufacturer's instructions. Once RNA concentration was quantified for each sample, cDNA was prepared using the SuperScript III First-Strand synthesis kit (Thermo Fisher Scientific). Subsequently, quantitative PCR (qPCR) was conducted using Fast SyBr green mix (Thermo Fisher Scientific) with sequence-specific primers for the targets as per the manufacturer's instructions. RNA expression of target genes was normalized to Cyclophilin A expression for mouse samples. Fast SyBr green was used for all qPCR reactions. One control was set to 1 and all other controls and test samples were normalized to that sample. Primer sequences are included in Supplementary Information.

## Semi-quantitative PCR for Srsf5 alternative splicing

As before, cells were processed for RNA isolation using the RNeasy kit (Qiagen) as per the manufacturer's instructions and cDNA was prepared using the SuperScript III First Strand synthesis kit (Thermo

Fisher Scientific) after RNA concentration quantification. Semi-quantitative PCR was carried out using GoTaq® DNA Polymerase (GoTaq Green Master Mix, Promega), with an annealing temperature of 56 °C for 40 cycles. The primer sequences used for the amplifications are included in Supplementary Information. The amplicons were resolved on 1.5% agarose gels, imaged at non-saturating levels, and image intensity was quantified using Adobe Photoshop.

### RNA-seq sample preparation for mouse and human cells

For mouse samples, islets were isolated from the indicated animals, and RNA isolation was carried out using the RNeasy kit (Qiagen) as per manufacturer's instructions. Libraries were prepared using the KAPA Stranded RNA-Seq Kits (Roche) as per the manufacturer's instructions. The libraries were sequenced on the Illumina HiSeq 4000. For the human samples, DOX treated and untreated eBCs were dissociated into single cells using accumax. Following dissociation, only live cells were isolated by flow sorting for further processing in the 10x genomics platform. The bar-coded libraries from the 10x platform were sequenced on Illumina Novaseq 6000.

### Western blotting

For westerns, human stem cell-derived clusters treated with or without DOX were homogenized in RIPA buffer and resolved on SDS-PAGE. Primary antibodies used were rabbit anti-SOX9 (Sigma), 1:1000, and mouse anti-Gapdh, 1:5000 (#sc-32233, Santa Cruz). Secondary antibodies were anti-rabbit IRDye 800CW (#827-08365, Odyssey) and anti-mouse IRDye 680LT (#827-11080, Odyssey).

### RNA-seq quantification and differential expression analysis

The raw fastq files were adapter-trimmed with cutadapt v1.13 and were subsequently processed with kallisto version 0.46.1[75] using the Ensembl GRCm38 as the reference transcriptome for building the index. The kallisto quantification was bootstrapped 100 times (using the -b parameter) and the output was processed for differential expression analysis using the sleuth package in R. We first performed a gene-level Wald test (gene_mode=True) between Sox9 knockout (KO) and control (WT) by aggregating the estimated isoform counts of each gene to their corresponding ensembl gene id in each condition. This test is however only able to identify over-expression or down-regulation in the overall gene expression between the two conditions. In order to further capture alternative splicing and isoform switching events we performed a transcript-level Wald test and aggregated the resulting p-values to the gene-level (aggregation_column='ens_gene'). This approach uses the Lancaster method[75] to compute a single p-value for each gene taking into account the individual p-values of its isoforms. In both gene- and transcript-level analyses, the raw p-values were used to compute Benjamini-Hochberg-adjusted false discovery rates (q values) and significance was set at 5% FDR throughout the manuscript.

### Single cell RNA-seq and TCC analysis

The raw fastq files for both samples (DOX/NO DOX) were pseudoaligned to the Ensembl GRCh38 cdna reference transcriptome using the kallisto bus command (version 0.46.1) and the transcript compatibility counts (TCCs) were generated with bustools (version 0.40.0)[76], filtering out reads that were multi-mapping across genes. Gene counts were also generated by aggregating TCCs to genes using bustools with the --genecounts flag. We subsequently filtered out 1) low quality cells based on their total UMI counts (60,848 cells kept) and 2) rare TCCs that were either detected in less than 1% of the cells or had a mean expression of less than 0.01 (61,006 equivalence classes kept). Doublets were removed based on the gene count data using Scrublet[77] with the default parameters. We then used scanpy[78] for downstream clustering and visualization. Gene counts were log-normalized, highly variable genes were selected with scanpy's pp.highly_variable_genes

function (flavor='seurat')[79] with the default parameters (2410 genes kept) and both total UMI counts and mitochondrial count percentages were regressed out. UMAP dimensionality reduction[80] was performed on the nearest neighbor graph computed with bbknn[81] on the PCA-reduced gene expression matrix (50 principal components). Clustering was performed with the Leiden algorithm implemented within scanpy with resolution parameter set to 0.2 and 0.5 to generate coarse and fine-grained resolution cluster labels. Cell types were annotated based on known gene markers and genes identified from a standard differential expression analysis with sc.tl.rank_genes_groups with the default parameters. The resulting gene lists were first filtered based on their p values (<10e-5) and then sorted based on their log-fold change. We then performed a high-dimensional TCC differential expression test between DOX and NO DOX treated beta-cells, including all TCCs mapping to SRSF5 with mean expression above 0.05 in both conditions[59]. The resulting raw p-value was extremely small (2e-50) which even after Bonferroni correction across all genes would indicate a significant change in the splicing patterns of SRSF5 in DOX-treated eBCs. For reference, testing for gene-level expression changes in SRSF5 between the same groups results in a raw *p*-value of 0.99.

### rMATS analysis

The raw fastq files were adapter-trimmed with cutadapt v1.13 and then mapped to the Ensembl reference genome GRCm38 using STAR v.2.7.3a[82]. To identify alternative splicing events, we used rMATS v4.0.2[46] on the indexed bam files with the default parameters and we visualized the resulting splicing events using ggsashimi[83].

### Statistical analyses

For all pairwise comparisons of gene expression, and hormone secretion in vivo and in vitro, Student's t-tests were performed to calculate the p values, and error bars represent standard error of the mean (S.E.M.) unless otherwise noted. Calculations for the p values for the mouse and human deep sequencing are described above in the Methods section. For alternative splicing, p values were generated by sleuth or rMATs as per the software parameters. The p value for the Enrichr is computed from the Fisher Exact test which is a proportion test that assumes a binomial distribution and independence for the probability of any gene belonging to any set.

### Reporting summary

Further information on research design is available in the Nature Portfolio Reporting Summary linked to this article.

## Data availability

The raw and processed sequencing data (bulk and single cell RNA-seq) generated in this study have been deposited in the Gene Expression Omnibus (GEO) database under accession number GSE245406. Source data are provided with this paper. Any additional data that support the findings of this study are available from the corresponding author upon request. Source data are provided with this paper.

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

## Acknowledgements

We would like to thank members of the Hebrok laboratory for helpful discussions. Imaging and flow cytometry experiments were supported by resources from the UCSF Diabetes and Endocrinology Research Center (DRC) and UCSF Flow Cytometry Core (NIH Diabetes Research Center grant P30 DK063720). G.N. was supported by a Kraft Family Fellowship and a JDRF postdoctoral fellowship (1-PNF-2016-320-S-B). Schematics in Figs. 1 & 4 were created with Biorender.com.

## Author contributions

S.P. and M.H. conceived the study. S.P. designed and conducted the experiments and wrote the paper. H.A.R. and G.N. designed and conducted experiments and edited the paper. H.M. conducted experiments and edited the paper. R.C. and P.P. conducted experiments; Z.C. performed data analysis; V.N. conducted data analysis, edited the paper, and participated in the interpretation of the data. H.A.R., G.N., and M.H. participated in discussions during the design of the study and the interpretation of the data.

## Funding

## Competing interests

MH holds stocks in Encellin Inc. and Thymmune Therapeutics Inc. and has received research support from Eli Lilly. He consults for Boxer Capital and CV Next and holds stock in the latter. He is the co-founder of Minutia Inc. and holds stocks and options in the company. The remaining authors declare no competing interests.
