## [Peer Review File · Nature Communications]

Sox9 regulates alternative splicing and pancreatic beta cell functionREVIEWER COMMENTS

Reviewer #1 (Remarks to the Author):

During development, SOX9 is essential in the pancreatic progenitor pool and in the differentiation of endocrine progenitors; however, while it has been reported to be expressed in the adult beta cell, by these authors and others, the role of Sox9 within insulin producing cells has not been previously explored. This manuscript from Puri and colleagues aims to understand what the role is for SOX9 within insulin-producing pancreatic β -cells. The authors knock out Sox9 in both recently-specified and mature mouse beta cells and show that the animals develop glucose intolerance but do not have marked changes in β -cell differentiation, maturation or gene expression. The authors hypothesize that they do not detect huge changes in gene expression because Sox9 has a unique role in regulating RNA splicing. They demonstrate that Sox9 loss leads to a change in RNA splicing that reduces protein-coding gene expression while simultaneously increasing non-coding gene expression. Finally, the authors replicate their studies in mice using hESC-derived SOX9 knock-out β -cells where they also show that SOX9 loss has relatively minor effects but splicing pathways are also dysregulated in this model. In sum, this is an interesting study that demonstrates that Sox9 deficient β -cells have altered splicing, which may lead to cell dysfunction and defects in metabolic homeostasis.

Concerns:

- 1) While there are a lot of data to support a role for Sox9 in the beta cell, the authors do not show that beta cells express Sox9 anywhere in this work. If antibodies are not sufficiently specific for this, perhaps the authors could use an RNA-based approach like RNAscope or use sorted β -cells?
- 2) Related, PMID 29326366 demonstrated SOX9 immunoreactivity in alpha cells; is it possible that SOX9 is expressed in other endocrine cell types found within the islet and this underlies some of the changes observed in the stem cell models? It would be useful to have T-SNE/UMAP plots presented for SOX9 and SRSF5 in SFig5.
- 3) It was not clear to me what the authors proposed as a mechanism that links SOX9 to SRSF5. Is it known how loss of SOX9 alters splicing of SRSF5, does a direct interaction of SOX9 with the spliceosome mediate this in beta cells or is this a secondary effect driven by gain or loss of another factor? At minimum, some discussion of how SOX9 alters beta cell splicing seems warranted.
- 4) Related to 3, the authors state that deregulated splicing occurs in stressed human beta cell lines and in T2D. Is it possible that the splicing defects are a secondary effect of loss of SOX9, which results in altered stress response? The induction of Fos, Jun, JunB, Egr1 etc seems to support such a hypothesis. To address this the authors may wish to carry out similar analyses on other knockout datasets presented in figure 2F. Is dysregulated splicing a common observation or is it specific to manipulation of Sox9 expression?
- 5) I could not figure out which InsCre transgene was used. Please cite original publications and describe in more detail so this work can be repeated

- 6) I could not figure out what age mice were used for the ITTs. Is this at a time when they show glucose intolerance or before that? Suggest the authors include age ranges for all animal figures, including RNA-expression analyses.
- 7) For the MIP-CreER studies, Cre controls are rather essential as they express growth hormone, which has been shown to alter beta cell function (PMID 26153246). This could also be the case for the InsCre mice (see 5) above.
- 8) Suggest the authors standardize their figures, and show all data points on all bar charts
- 9) I found the dysregulated insulin processing interesting. Any idea why knockout mice had elevated proinsulin levels? Were processing enzymes differentially expressed?
- 10) It wasn't clear to me why the authors did not wait until β -cell differentiation was complete before they treated their eBCs with Dox. It seems to me like this would be the "cleanest" way to show that the phenotypes observed were not due to changes in differentiation?
- 11) Related, were control differentiations carried out in the presence of Dox; does dox cause cell stress that could impact splicing, e.g. PMIDs 29184058, 34183648, 32152409?

Reviewer #2 (Remarks to the Author):

To authors,

In this manuscript, the authors mainly focused on the role of Sox9 in mature pancreatic beta cells and found that Sox9 continued to play important roles in regulating function within mature pancreatic beta cells despite of its low expression level. Their studies reported that loss of Sox9 may affect insulin secretion in beta cells through regulating the splicing of important splicing regulators. In addition, they also evaluated the role of Sox9 in human islets and found Sox9 playing a conserved role in both rodent and human beta cells. Although deletion of Sox9 had little effect on endocrine lineage, removal of Sox9 led to blunted insulin secretion in human stem cell derived beta cells. The phenotype of Sox9 KO mice is interesting and has extended our current knowledge about the role of Sox9 in beta cell dysfunction and related disease. However, this study is just descriptive and lack the mechanism by which Sox9 deletion affected beta cell function. In addition, their data can not fully support the conclusion that loss of Sox9 affected insulin secretion mainly through regulating the splicing of Srsf5.

Major concerns,

1. The authors did not provide persuasive data to support that alternative splicing is responsible for the impaired insulin secretion observed with Sox9 deletion. Did the authors detect the splicing of crucial regulators involved in insulin secretion? For example, they mentioned that loss of Sox9 decreased Glut2 protein expression without affecting its mRNA level. Did they detect whether the splicing of Glut2 was also affected with Sox9 deletion?

2. From the data provided in the manuscript, we can only conclude that loss of Sox9 may lead to alternative splicing of Srsf5. However, alternative splicing of Srsf5 observed could also be a secondary phenotype caused by alternation of other regulators. The authors did not provide data to fully elucidated that loss of Sox9 exerted direct effect on RNA splicing.

3. Since the authors emphasis on evaluating the role of Sox9 in mature beta cells. They should perform more experiments on MIP-CreERTSox9^{-/-} mice, for example, the in vivo insulin secretion which is crucial for their conclusions. And it would be more reasonable to perform bulk RNA sequencing with islets from MIP-CreERT;Sox9^{-/-} mice instead of using the islets from Ins-Cre;Sox9^{-/-} mice.

4. The authors mentioned that Sox9 knockout mice exhibited glucose intolerance as they aged, which can mimic the progressive loss of glucose homeostasis during diabetes. So, did the authors detect the expression alternation of Sox9 in mice under certain metabolic stress, such as aging or HFD-feeding?

5. From the single cell RNA-seq result, the authors found increased cell accumulation in alpha and delta-like populations with Sox9 depletion, on the contrary, they did not observe an increase in GCG-expression cells by FACS analysis with Sox9 removal. The authors believe that the discrepancy between transcript level and functional protein accumulation upon loss of Sox9 can be explained by the non-functional transcripts which may not be translated. However, this explanation seems illogical. It implies that even loss of Sox9 may affect RNA splicing, it will not result in the alternation of protein expression.

6. It is difficult to agree with the authors that the insulin was “only moderately reduced” in Sox9 KO mice. As shown in Fig.2A, the insulin was markedly decreased while the proinsulin was significantly increased with Sox9 KO, therefore, the authors should focus on how loss of Sox9 impaired the processing of the precursor into the mature hormone peptide.

Minor concerns,

1. The authors should detect protein abundance of Srsf5 upon Sox9 loss.

2. Why no second phase of insulin secretion can be observed even in the control group (Fig. 4F)?

3. Some description in the manuscript is not so accurate. For example, in Fig.2A, the membrane translocation of Glut2 was significantly decreased with Sox9 deletion, but we can hardly identify the alternation of Glut2 protein abundance from the immunofluorescent figures, western bolt can be carried out to quantity the protein expression of Glut2.

4. Although it has no statistical difference, the insulin secretion was also increased in Sox9 KO mice after glucose stimulation (Fig.1D). Given that the sampling variability is high in each group, the authors should enlarge the sample size in this test.

5. It is unreasonable that there is no SEM error bar for the time point of 0 min (Fig.S1C,the Y-axis denotes the blood glucose level), especially there are only 3 mice per group.

6. The authors should check and uniform the format of all the references.

Reviewer #3 (Remarks to the Author):

Puri et al. used a comprehensive set of experiments to explore a previously unanticipated role for Sox9, a transcriptional regulator of embryonic pancreas and endocrine cell development, in mature beta cells. Overall, the experiments were well-designed and the manuscript is clearly written. The presented results are interesting and useful for better understanding mature beta cell functions. However, the following issues need to be addressed to better support the conclusions and claims.

Figure 1A-E: Did the authors observe any sex-dependent effect in comparisons presented in these panels?

Line 78: What are the considerations to delete exons 2 and 3 from Sox9, in order to eliminate its expression?

Line 151: In the gene ontology analysis, were the p values adjusted for multiple testing?

Line 211: The authors need to provide a more detailed explanation about data presented in Figure 3A. The main text says there are 315 identified transcripts but Figure 3A shows 315 genes. If a transcript is differentially expressed between two conditions, this is not equivalent to the corresponding gene being differentially spliced. A counter example is when a gene's transcripts all have decreased expression by 50% in one condition compared with another. In this case, the gene is not differentially spliced, but all transcripts are differentially expressed. However, the authors are using the two concepts interchangeably.

Figure 3D: What are the number of splicing events in each category?

Line 243: The manuscript shows that both sleuth and rMATS have detected alternative splicing in the gene Srsf5. How about the other genes listed in Fig 3B? Were they also detected by both methods? Which exons were alternatively spliced and what was the magnitude of the changes as indicated by the methods?

For Srsf5, it would be helpful to add a plot showing the specific exon compositions of its nine isoforms?

Line 339: A comparison of proportions of different cell populations in control and SOX9-depleted conditions can be added. It would be interesting to see if any populations are enriched in control or SOX9-depleted samples. Fig 5D is helpful, but a side-by-side boxplot with statistical significance might be more straightforward.

Figure 5E: Similar as commented above, it would be helpful to add a graph to explain the structural difference between the equivalent classes.

Wherever applicable, a more detailed description of how the p values were obtained is needed in the Figure captions or Methods section.

The data is not uploaded to a public repository and the data analysis scripts/code are not available. This prevents other researchers from reproducing or following up with the study.

Minor:

Figure 1: What's the definition of relative transcript level?

Figure 2C: Please add a description of the expression levels in the caption or legend.

REVIEWER COMMENTS

We would like to thank the reviewers for their thorough review of our original manuscript, the supportive assessment of the importance of our findings, and the thoughtful comments raised to further enhance the impact of our findings. We have addressed all comments in detail below and have marked our responses in blue.

Reviewer #1 (Remarks to the Author):

During development, SOX9 is essential in the pancreatic progenitor pool and in the differentiation of endocrine progenitors; however, while it has been reported to be expressed in the adult beta cell, by these authors and others, the role of Sox9 within insulin producing cells has not been previously explored. This manuscript from Puri and colleagues aims to understand what the role is for SOX9 within insulin-producing pancreatic β -cells. The authors knock out Sox9 in both recently-specified and mature mouse beta cells and show that the animals develop glucose intolerance but do not have marked changes in β -cell differentiation, maturation or gene expression. The authors hypothesize that they do not detect huge changes in gene expression because Sox9 has a unique role in regulating RNA splicing. They demonstrate that Sox9 loss leads to a change in RNA splicing that reduces protein-coding gene expression while simultaneously increasing non-coding gene expression. Finally, the authors replicate their studies in mice using hESC-derived SOX9 knock-out β -cells where they also show that SOX9 loss has relatively minor effects but splicing pathways are also dysregulated in this model. In sum, this is an interesting study that demonstrates that Sox9 deficient β -cells have altered splicing, which may lead to cell dysfunction and defects in metabolic homeostasis.

We appreciate the overall assessment and positive comments by the reviewer. We have addressed all comments raised, paying special attention to visualizing the expression of SOX9 in the pancreatic beta cells. We believe that our new data, in combination with the functional consequences of Sox9 deletion using a beta cell-specific Cre recombinase shown in our original submission, clearly demonstrate a pivotal and novel role for the transcription factor in regulating beta cell function.

Concerns:

1) While there are a lot of data to support a role for Sox9 in the beta cell, the authors do not show that beta cells express Sox9 anywhere in this work. If antibodies are not sufficiently specific for this, perhaps the authors could use an RNA-based approach like RNAScope or use sorted β -cells?

We fully agree with the reviewer that showing Sox9 expression in beta cells would elevate the impact of our findings and alleviate concerns raised by this reviewer and others. We have tested several antibodies to show the presence of the protein in the nuclear fraction, and have got variable results that could be attributed to either quality of antibody or the levels of the protein that may be too low to be detected reliably in this cell type. In response to the reviewer's comment, we spent an extended period of time to establish the RNAScope technology in our lab. We now include exciting data from the RNAScope analysis demonstrating that Sox9 is expressed in the nuclei of control islets. Importantly, the signal is absent in insulin-positive cells in *Ins-Cre*; *Sox9*^{-/-} animals. To our knowledge, this is the first evidence of Sox9 expression in insulin-producing cells. We greatly appreciate the suggestion by the reviewer to test this technique as a method to detect a low expressing gene, and are confident that this information will generate excitement in the beta cell community as it challenges a long established dogma that Sox9 expression and function is confined to pancreas progenitors and exocrine duct cells. We have included these novel and significant findings in Fig. 1B of the revised manuscript.

2) Related, PMID 29326366 demonstrated SOX9 immunoreactivity in alpha cells; is it possible that SOX9 is expressed in other endocrine cell types found within the islet and this underlies some of the changes observed in the stem cell models? It would be useful to have T-SNE/UMAP plots presented for SOX9 and SRSF5 in SFig5.

Our RNAScope analyses clearly show positive nuclear accumulation of Sox9 probe in non-beta islet cells, demonstrating the presence of this protein in other endocrine cell types. We believe that the alternative splicing regulated by SOX9 is a conserved phenomenon, and it may be taking place in other cells along with those that express insulin. The overlapping phenotypes that we report between the rodent and human models of SOX9 deletion, however, convince us that the changes stem from the beta cell. As the reviewer suggested, we have included UMAP plots for SOX9 and SRSF5 in the revised Fig. S5. The UMAP plots show that SOX9 is expressed at highest levels in population 6, co-expressed with HES1, GMNN, and CCND1, marking the

possible ductal fated cells, as expected; the next most abundant population is the beta cell progenitors, also unsurprisingly as this population likely represents early endocrine progenitors known to express SOX9. Expression of SOX9 is low across all other populations, supporting the previous studies that the usual methods of detection for this protein have not been successful in quantifying the abundance of this candidate in islet cells. SRSF5, on the other hand, is expressed ubiquitously across all populations, as would be expected from a protein that participates in splicing.

3) It was not clear to me what the authors proposed as a mechanism that links SOX9 to SRSF5. Is it known how loss of SOX9 alters splicing of SRSF5, does a direct interaction of SOX9 with the spliceosome mediate this in beta cells or is this a secondary effect driven by gain or loss of another factor? At minimum, some discussion of how SOX9 alters beta cell splicing seems warranted.

This is an important question. There is prior evidence from just a few studies that describe direct interaction of Sox9 with splicing factors. Sox9 is predicted to modify splicing patterns by its association with Y14, an RNA-binding protein that forms part of the exon junction complex and has also been associated with ribonucleoprotein transcripts in an RNA-dependent manner, colocalizing with SR proteins (Girardot et al., 2018; Penrad-Mobayed et al., 2018). Unfortunately, the very low level of Sox9 protein below detection has so far prevented us from performing the required co-IP assays to demonstrate that similar interactions exist in beta cells. We appreciate the reviewer's comments, and have expanded the discussion of the potential mechanisms of splicing in our revised manuscript.

4) Related to 3, the authors state that deregulated splicing occurs in stressed human beta cell lines and in T2D. Is it possible that the splicing defects are a secondary effect of loss of SOX9, which results in altered stress response? The induction of Fos, Jun, JunB, Egr1 etc seems to support such a hypothesis. To address this the authors may wish to carry out similar analyses on other knockout datasets presented in figure 2F. Is dysregulated splicing a common observation or is it specific to manipulation of Sox9 expression?

We agree with the reviewer that the exact mechanism that SOX9 uses to alter the splicing within beta cells still need to be understood. It is likely that there are direct AND indirect mechanisms at play; analyses of the datasets presented in figure 2F may still not be able to discern between these scenarios. The Enrichr analysis shows significant overlap between genes that are dysregulated upon Sox9 knockout in beta cells in our hands and another dataset with genes that are downstream of Sox9, underscoring the conserved nature of these candidates as Sox9 targets. However, as stated in response to comment #3, there is growing evidence that Sox9 directly interacts with splicing factors. While the low level of Sox9 protein currently prevents a direct interrogation of such questions, we hope to address this issue via genetic means in the future.

5) I could not figure out which InsCre transgene was used. Please cite original publications and describe in more detail so this work can be repeated

We apologize for this oversight – we have now included the original reference of the study that generated this mouse model in the Methods section of the manuscript.

6) I could not figure out what age mice were used for the ITTs. Is this at a time when they show glucose intolerance or before that? Suggest the authors include age ranges for all animal figures, including RNA-expression analyses.

We have now added the age of the animals for the ITT in the figure and the figure legend as well as the Methods section. We have further included such information for all animal figures either in the figure itself, in the figure legend, or in the Methods section. We thank the reviewer for bringing this to our attention.

7) For the MIP-CreER studies, Cre controls are rather essential as they express growth hormone, which has been shown to alter beta cell function (PMID 26153246). This could also be the case for the InsCre mice see 5) above.

We appreciate the comment regarding the growth hormone and how it might influence metabolic outcomes in animals. Due to our mating scheme, we were unable to generate Cre alone animals that were wild type for both alleles of Sox9. However, we do have data using the Ins-Cre (some of which are included in the manuscript) that demonstrate that a loss of one allele of Sox9 only triggers a partial phenotype (Fig. 1A, blue curve, Fig S1B). If indeed the predominant phenotype was Cre dependent, we would anticipate the Cre-

positive animals to demonstrate an overlapping phenotype with the Sox9-knockout cohort, and we do not observe that. The haplo-insufficient phenotype we document here is in fact well known to occur in other Sox9 depletion models (for e.g. work by Maïke Sander).

8) Suggest the authors standardize their figures, and show all data points on all bar charts

We have incorporated the reviewer's suggestion and standardized our bar graphs by showing the individual points wherever appropriate in the primary figures within the manuscript.

9) I found the dysregulated insulin processing interesting. Any idea why knockout mice had elevated proinsulin levels? Were processing enzymes differentially expressed?

The reviewer raises an interesting question. We anticipated that the increased accumulation of proinsulin upon Sox9 knockout might be caused by changes in the activity of the processing enzymes. However, we could not find significant changes in the RNA levels of Pcsk1 and Pcsk2 in the deep sequencing data set, even though there was a mild downregulation in the transcript levels (data not shown). When we looked at the alternative splicing of these genes, we also did not find any altered accumulation of splicing isoforms.

In addition to mature insulin processing, there is a large body of work that has interrogated the role of proinsulin misfolding in the ER as a contributor to beta cell dysfunction, including work from Peter Arvan and others. Our data support this notion and point to erroneous processing of several ER chaperones, including Hsp90ab1, Hsph1, Syvn1, Ddit3, and Xbp1 (a key regulator of the ER-associated degradative pathway) among others (Fig. S3B). Thus, while we could not detect defects in insulin processing enzymes, our data indicate that Sox9 loss perturbs ER stress responses known to control proinsulin folding.

10) It wasn't clear to me why the authors did not wait until β -cell differentiation was complete before they treated their eBCs with Dox. It seems to me like this would be the "cleanest" way to show that the phenotypes observed were not due to changes in differentiation?

Although we have been able to generate eBCs that are highly similar to primary human islets, these clusters do not persist past 1 week in culture after formation. In fact, longer incubations lead to dedifferentiation of the clusters and loss of insulin expression, similar to what is known to occur with primary islets as well. Therefore, we added Dox at an earlier time point to delete SOX9 and query the effects in the resulting eBCs.

11) Related, were control differentiations carried out in the presence of Dox; does dox cause cell stress that could impact splicing, e.g. PMIDs 29184058, 34183648, 32152409?

We appreciate the reviewer's comment – we have indeed carried out experiments where control (unedited) human stem cells have been treated with dox during differentiation, and our initial analyses do not reveal any adverse effects of the dox dose that we have been using in our experiments.

As we show in the data below (Reviewer 1, Fig. 1), the differentiation of the unedited INS-GFP line treated with dox is similar to the differentiation without dox treatment, ascertained by the total numbers that are generated after dox treatment, and the total percent of GFP-positive cells that are collected at the end of the differentiation protocol. We are currently including these data in a different manuscript; however, we share it here with the reviewer to demonstrate our findings that dox treatment alone does not perturb the yield of insulin-expressing cells. Considering that our differentiations are

Reviewer 1, Fig. 1: Quantification of total cell counts and GFP-positive cells upon addition of doxycycline during differentiation of human stem cells into insulin-producing cell

not affected by the dox levels uses, we did not feel it warranted extensive deep sequencing analyses on these cells.

Reviewer #2 (Remarks to the Author):

To authors,

In this manuscript, the authors mainly focused on the role of Sox9 in mature pancreatic beta cells and found that Sox9 continued to play important roles in regulating function within mature pancreatic beta cells despite of its low expression level. Their studies reported that loss of Sox9 may affect insulin secretion in beta cells through regulating the splicing of important splicing regulators. In addition, they also evaluated the role of Sox9 in human islets and found Sox9 playing a conserved role in both rodent and human beta cells. Although deletion of Sox9 had little effect on endocrine lineage, removal of Sox9 led to blunted insulin secretion in human stem cell derived beta cells. The phenotype of Sox9 KO mice is interesting and has extended our current knowledge about the role of Sox9 in beta cell dysfunction and related disease. However, this study is just descriptive and lack the mechanism by which Sox9 deletion affected beta cell function. In addition, their data cannot fully support the conclusion that loss of Sox9 affected insulin secretion mainly through regulating the splicing of Srsf5.

We would like to thank the Reviewer for their positive assessment of our findings and the constructive questions. We have addressed the comments raised in detail below. We do want to emphasize that the study goes beyond just a descriptive analysis. In fact, we feel our study is paradigm shifting in that it identifies Sox9, a gene previously only found in pancreas progenitor and mature duct cells, as a novel regulator of alternative splicing in rodent and human beta cells. While illuminating the exact mechanism by which Sox9 controls alternative splicing events is currently technically not feasible due to the low expression of Sox9 protein that currently prevents such studies in beta cells, we do provide ample information that Sox9 deletion profoundly changes the activity of splicing enhancers at the top of the splicing hierarchy. Increasing evidence points to alternative splicing as an emerging key regulator of beta cell function with connections to diabetes. A recent manuscript focusing on SRp55 (SRSF6), a gene with splicing defects in beta cells lacking Sox9 in our study, links its dysfunction to defects in insulin release (Juan-Mateu et al, Eizirik, Diabetes 2018;67:423–436). Thus, our study provides critical new information to support the notion of alternative splicing as an essential regulator of overall beta cell function.

Major concerns:

1. The authors did not provide persuasive data to support that alternative splicing is responsible for the impaired insulin secretion observed with Sox9 deletion. Did the authors detect the splicing of crucial regulators involved in insulin secretion? For example, they mentioned that loss of Sox9 decreased Glut2 protein expression without affecting its mRNA level. Did they detect whether the splicing of Glut2 was also affect with Sox9 deletion?

The reviewer raises an important question, namely what are the contributions of Sox9 as a transcription factor versus its effects as regulator of alternative splicing on beta cell function. Our data indicate that while the effects of Sox9 on overall gene expression are dwarfed by its role regulating alternative splicing, we do observe clear transcriptional changes in pathways related to beta cell function. For example, as shown in Figure 2D, the KEGG pathway analysis revealed changes in 'Insulin resistance', 'MAPK signaling', and 'Maturity onset of diabetes of the young' pathways. KEGG pathway analysis reveals 'Insulin Resistance' and the genes populating this pathway include Irs1, Irs2, Ppp1r3c, among others (Table S5). Irs1 and Irs2 have been implicated in type 2 diabetes – mechanistically, these proteins are sensors for ER stress (working through Xbp1 splicing) to cause beta cell dysfunction (Takatani et al., 2016).

With regards to alternative splicing, we favor the model that Sox9 loss causes broad changes in splicing activity, rather than specific defects in splicing of only select few factors involved in insulin secretion. For example, in our analyses of the deep sequencing data from mouse, we did not uncover any significant change in the Glut2 isoform levels in cells lacking Sox9. Thus, it does not appear that defective splicing causes the change in Glut2 localization and possibly function. Changes in Glut2 have been observed in several models of metabolic dysfunction and might be a consequence of the perturbations within the overall beta cell machinery. Furthermore, we did not find significant alternative splicing defects of other regulators of insulin secretion, including Kir6.2, Sur1, ZnT8, Pcsk1, and Pcsk2.

On the other hand, Sox9 deletion does cause defects in alternative splicing in key factors of other pathways. Importantly, Xbp1 is one of the regulators that appear on our list of genes with altered alternative splicing (Fig. S3B), strongly supporting the hypothesis of aberrant protein processing within the secretory pathway. Furthermore, Txnip, another ER stress protein is alternatively spliced as well. We also demonstrate that Sox9 deletion switches alternative splicing patterns in Srsf5 and Srsf6, proteins that serve as enhancers of alternative splicing, to prevent efficient generation of their functional proteins. In particular, Srsf6 silencing has previously been shown to impair insulin secretion (Juan-Mateu et al, Eizirik, Diabetes 2018;67:423–436).

Summarily, our current working hypothesis is that the global changes that occur within Sox9-depleted cells are working in concert to dampen the glucose sensing and therefore the secretory response in beta cells. As stated above, the findings that Xbp1 splicing and Txnip expression are altered point strongly to a perturbation in the ER processing mechanisms, defects that are strongly linked to the development of dysglycemia and diabetic phenotypes.

2. From the data provided in the manuscript, we can only conclude that loss of Sox9 may lead to alternative splicing of Srsf5. However, alternative splicing of Srsf5 observed could also be a secondary phenotype caused by alternation of other regulators. The authors did not provide data to fully elucidated that loss of Sox9 exerted direct effect on RNA splicing.

As already discussed in response to question #3 for Reviewer 1, we agree that this is an important question. There is growing evidence that Sox9 can directly interact with RNA binding and splicing factors, thus suggesting a direct role in regulating alternative splicing. However, we cannot rule out indirect effects and due to the very low level of Sox9 protein in beta cells are currently unable to assess as to whether the factor directly binds to splicing components. We have expanded our manuscript and discussed these possibilities in the revision. We plan to address these questions using genetic means in future studies.

3. Since the authors emphasis on evaluating the role of Sox9 in mature beta cells. They should perform more experiments on MIP-CreERTSox9^{-/-} mice, for example, the *in vivo* insulin secretion which is crucial for their conclusions. And it would be more reasonable to perform bulk RNA sequencing with islets from MIP-CreERT;Sox9^{-/-} mice instead of using the islets from Ins-Cre;Sox9^{-/-} mice.

We agree with the reviewer that a deeper analysis of the islets isolated from the adult deletion of Sox9 in an animal model would be informative – unfortunately, due to a crisis in mating in the MIP-CreERT;Sox9^{-/-} colony (as a result of Covid restrictions), we do not have those mice any more and cannot carry out that exact experiment. In lieu of the studies suggested by the reviewer, we adopted a different approach to answer the same question. We isolated islets from adult Sox9^{fl/fl} mice, added Adeno-Cre, and carried out hormone secretion analyses on these islets. These data are now shown in Fig. 1J&K in the revised manuscript. Our conclusions from these experiments are that in adult islets, upon Cre-based deletion of Sox9, we can induce changes that lead to a stunting of beta cell function when challenged with high levels of glucose. These data confirm what we have

Reviewer 2, Fig. 1: RNAScope analysis of Sox9 expression in animals under conditions of metabolic stress.

seen in several models – the *in vivo* models with the *Ins-Cre* and the *MIP-Cre^{ERT}* driven loss of Sox9, and the *in vitro* human stem cell derived models – all pointing to a pivotal role of Sox9 in regulating beta cell function.

4. The authors mentioned that Sox9 knockout mice exhibited glucose intolerance as they aged, which can mimic the progressive loss of glucose homeostasis during diabetes. So, did the authors detect the expression alternation of Sox9 in mice under certain metabolic stress, such as aging or HFD-feeding?

Based on the reviewer’s comment, we decided to look for the expression of Sox9 under conditions of metabolic stress. Although in the past immunodetection of Sox9 has been difficult, we have successfully established RNAScope in our lab to detect Sox9 RNA in the islets (Fig. 1B in revised manuscript). Therefore, we used this technique on tissues isolated from mice that were fed for prolonged periods on either normal chow or a high fat diet. The data are included here (Reviewer 2, Fig. 1). We found that in animals fed with normal chow, we could still detect Sox9 in beta cells of rodent islets. However, animals that had been treated with a high fat diet for greater than one year had significantly reduced accumulation of Sox9 in the islets. These data suggest that metabolic stress due to an altered high fat diet reduces Sox9 expression. We would be glad to include these novel data in the revised manuscript if the reviewer deems it necessary.

5. From the single cell RNA-seq result, the authors found increased cell accumulation in alpha and delta-like populations with Sox9 depletion, on the contrary, they did not observe an increase in GCG-expression cells by FACS analysis with Sox9 removal. The authors believe that the discrepancy between transcript level and functional protein accumulation upon loss of Sox9 can be explained by the non-functional transcripts which may not be translated. However, this explanation seems illogical. It implies that even loss of Sox9 may affect RNA splicing, it will not result in the alternation of protein expression.

We apologize for not making this point clear in the original version of the manuscript. Our protein and transcript data show that the number of cells expressing GCG is not increasing but that the proportion of cells expressing mature beta cell markers is decreasing upon dox treatment. We calculated the proportions of high GCG expressing cells in both samples using a threshold of 4 log-normalized counts (Reviewer 2, Fig. 2). The threshold was chosen to separate the bimodal distribution of GCG expression across all cells (see attached figure). This approach directly queries the proportion of cells with a distinguishable expression of GCG and is more in line with what can be detected by FACS analysis. Overall, we found that high GCG expressing cell proportions are comparable between no dox and dox samples (6% and 6.5% respectively). We hope that we have addressed the reviewer’s concern with this additional information.

Reviewer 2, Fig. 2: Comparison of the GCG-positive populations without or with doxycycline addition during stem cell differentiation shown as UMAPs and histograms.

6. It is difficult to agree with the authors that the insulin was “only moderately

reduced” in Sox9 KO mice. As shown in Fig.2A, the insulin was markedly decreased while the proinsulin was significantly increased with Sox9 KO, therefore, the authors should focus on how loss of Sox9 impaired the processing of the precursor into the mature hormone peptide.

We agree with the reviewer's concern that there likely is a processing defect in the cells depleted of Sox9. We are pursuing this line of inquiry as a follow up study to this one, and appreciate the comments of the reviewer. We have also changed the description of the decreased insulin expression in the manuscript to more accurately reflect the observed changes. As mentioned above, we believe that a processing defect is highly likely since the genes that are differentially alternatively spliced cluster together in the pathways of ER folding, including the unfolded protein response regulators Xbp1 and Txnip.

Minor concerns,

1. *The authors should detect protein abundance of Srsf5 upon Sox9 loss.*

As recommended by the reviewer, we tested for Srsf5 protein in control and knockout mouse tissue. As now shown in Fig. S3E, we can see a significant perturbation of the staining pattern for Srsf5 upon loss of Sox9. We observe a reduction in the nuclear signal upon Sox9 deletion – further strengthening our conclusion that although no change was detected on the gene level by deep sequencing, there is a shift in the protein accumulation in the cells that have lost Sox9 activity.

2. *Why no second phase of insulin secretion can be observed even in the control group (Fig. 4F)?*

It is well known in the field of stem cell to beta cell differentiation that current protocols are suboptimal in generating fully mature cells that have robust first and second phase insulin secretion. In fact, although several labs are currently attempting to improve this phenotype, most groups rely on the *in vivo* maturation that occurs post transplantation in small animal models. We appreciate the reviewer's comment – this is certainly a phenomenon we are trying to improve by tweaking our differentiation protocols and hope to solve this in the near future.

3. *Some description in the manuscript is not so accurate. For example, in Fig.2A, the membrane translocation of Glut2 was significantly decreased with Sox9 deletion, but we can hardly identify the alternation of Glut2 protein abundance from the immunofluorescent figures, western blot can be carried out to quantify the protein expression of Glut2.*

We appreciate the reviewer's comment. By qPCR analyses and by deep sequencing of isolated islets, we did not detect any changes in Glut2 expression; it is likely that the change in the localization of the protein reflects a change in the processing or functional capacity of the protein. Unfortunately, we currently have limiting material as we have utilized all the animals while addressing the comments in this revision and we have been unable to collect enough material to carry out a successful western blot for Glut2. We will keep this comment in mind as we continue our efforts to fully define the role of Sox9 in the pancreatic beta cell.

4. *Although it has no statistical difference, the insulin secretion was also increased in Sox9 KO mice after glucose stimulation (Fig. 1D). Given that the sampling variability is high in each group, the authors should enlarge the sample size in this test.*

At the time of our original submission, we were limited in the number of knockout animals in our colony – we have since then generated more animals, and as recommended by the reviewer, have repeated these analyses, with similar outcomes. The 'n' numbers have been increased in the revision, as seen in Fig. 1E&F. Our conclusions are strengthened by the new data – although Sox9-deleted animals can secrete insulin in response to glucose, this response is defective.

5. *It is unreasonable that there is no SEM error bar for the time point of 0 min (Fig.S1C, the Y-axis denotes the blood glucose level), especially there are only 3 mice per group.*

Due to variability between animals, we normalized all the values in each animal to the t=0 min time point. Consequently, all t=0 min time point are all 0, and there is no SEM bar.

6. *The authors should check and uniform the format of all the references.*

We have checked all the references and formatted them as suggested by the reviewers. Once the manuscript is accepted, we will ensure that the references are correctly formatted.

Reviewer #3 (Remarks to the Author):

Puri et al. used a comprehensive set of experiments to explore a previously unanticipated role for Sox9, a transcriptional regulator of embryonic pancreas and endocrine cell development, in mature beta cells. Overall, the experiments were well-designed and the manuscript is clearly written. The presented results are interesting and useful for better understanding mature beta cell functions. However, the following issues need to be addressed to better support the conclusions and claims.

We thank the Reviewer for their positive comments on the design of the study and the clear presentation of the manuscript. We have addressed all comments as detailed below.

Figure 1A-E: Did the authors observe any sex-dependent effect in comparisons presented in these panels?

We did address this question, but only in the Methods section. We did observe sex-dependent effects; females did not develop glucose intolerance and therefore were not included in the analyses. We would be glad to include this information in the main text if the Reviewer feels this enhances the impact of the study.

Line 78: What are the considerations to delete exons 2 and 3 from Sox9, in order to eliminate its expression?

Murine Sox9 is encoded by 3 exons, and as described in the original paper that generated the floxed allele (Kist et al., 2002), elimination of exons 2 and 3 removes half of the DNA-binding high mobility group (HMG) domain and the transactivation domain, effectively rendering the resulting product non-functional and thereby mimicking a complete knockout.

Line 151: In the gene ontology analysis, were the p values adjusted for multiple testing?

For the gene ontology analysis, multiple testing (including Bonferroni and Benjamini tests) was carried out and the values are shown in Supplemental Table S2-S5.

Line 211: The authors need to provide a more detailed explanation about data presented in Figure 3A. The main text says there are 315 identified transcripts but Figure 3A shows 315 genes. If a transcript is differentially expressed between two conditions, this is not equivalent to the corresponding gene being differentially spliced. A counter example is when a gene's transcripts all have decreased expression by 50% in one condition compared with another. In this case, the gene is not differentially spliced, but all transcripts are differentially expressed. However, the authors are using the two concepts interchangeably.

We completely agree with the reviewer that using genes and transcripts interchangeably is misleading and can cause confusion. We have now changed the text in the manuscript to reflect the correct nomenclature. We have identified 315 genes that are alternatively spliced in a significant way. We have edited the manuscript to incorporate this change, and thank the reviewer for ensuring that the correct terms are used.

Figure 3D: What are the number of splicing events in each category?

For Figure 3D, the total number of significant events were calculated to be 1587. The distribution of the events is shown below.

A3SS- 140

A5SS- 101

MXE- 93

RI- 466

SE- 787

All possible events in the mouse genome by default (shown in Figure S3C) total to 76,354, with the following distribution.

A3SS- 4746

A5SS- 2836

MXE- 10116

RI- 3105

In the revised manuscript, we have added the total numbers to the respective figures to provide further information for the readers.

Line 243: The manuscript shows that both sleuth and rMATS have detected alternative splicing in the gene Srsf5. How about the other genes listed in Fig 3B? Were they also detected by both methods? Which exons were alternatively spliced and what was the magnitude of the changes as indicated by the methods? For Srsf5, it would be helpful to add a plot showing the specific exon compositions of its nine isoforms?

14 out of the top 20 genes (Fig. 3B) are detected by rMATS, with an FDR value <0.05 , making them highly significant. We have included this finding in the revised manuscript. We have also included a table in the supplementary data (Table S12) that lists the genes and the splicing events associated with them that were identified by rMATS. We hope that these additions help address the reviewer's comments.

We have also added the exon compositions of the isoforms of Srsf5 to the revised Fig. 3B.

Line 339: A comparison of proportions of different cell populations in control and SOX9-depleted conditions can be added. It would be interesting to see if any populations are enriched in control or SOX9-depleted samples. Fig 5D is helpful, but a side-by-side boxplot with statistical significance might be more straightforward.

We appreciate the reviewer's comments. We re-plotted the data to represent proportions of populations within the different clusters, shown here in a side-by-side boxplot (Reviewer 3, Fig. 1). However, statistical significance cannot be calculated for the data as these are the results of one differentiation, and although the number of events are large, it is difficult to extract statistical information from such single cell deep sequencing data due to limited biological replicates. We are happy to include this boxplot in supplemental figure 5 if the reviewer feels that it improves the quality of the manuscript.

Reviewer 3, Fig. 1: Proportion of cell populations in the six identified clusters upon single cell sequencing of human stem cells differentiated through a 26-day protocol with or without doxycycline treatment.

Figure 5E: Similar as commented above, it would be helpful to add a graph to explain the structural difference between the equivalent classes.

By definition, equivalence classes are sets of reads that are multi-mapping to the same set of transcripts in a reference transcriptome [Ntranos et al., Genome Biology, 2016, Bray et al., Nature Biotechnology, 2016]. Although there is no direct way to visualize them as a linear segment on each individual transcript, one can think of equivalence classes as the union of all subsequences of a given read length that are common between the transcripts in the multi-mapping set. For example, in the case of SRSF5, equivalence class 118326 contains reads that map to the overlapping region of isoforms encoding non-functional SRSF5 proteins (isoforms 207, 211 and 212) and was the only class that was enriched in the DOX treated samples. All three isoforms have retained introns ('poison exon') that lead to either no protein or a transcript that is targeted for nonsense-mediated degradation, thereby further strengthening our analysis of non-functional transcripts of SRSF5 being upregulated upon SOX9 deletion. We hope that this explanation clarifies the reviewer's concern and we are happy to include it in the manuscript if the reviewer thinks it will be useful for the readers.

Wherever applicable, a more detailed description of how the p values were obtained is needed in the Figure captions or Methods section.

Upon the reviewer's suggestion, we have added a paragraph in the Methods section to list what methods were used to generate the p values for the various analyses included in the manuscript.

The data is not uploaded to a public repository and the data analysis scripts/code are not available. This prevents other researchers from reproducing or following up with the study.

We are excited to share our findings with the research community and will upload all deep sequencing data to a public repository as soon as the paper is accepted.

Minor:

Figure 1: What's the definition of relative transcript level?

Relative transcript level reflects the level of transcript as compared to a control group. After qPCR, ddCt values are calculated normalized to a housekeeping gene, and one of the control samples is measured as 1, with all other values normalized to that. This allows the quantification of variability between control samples, and also a comparison between the test samples as the control cohort.

Figure 2C: Please add a description of the expression levels in the caption or legend.

We thank the reviewer for pointing this out and have modified the figure legend.

REVIEWER COMMENTS

Reviewer #1 (Remarks to the Author):

The authors have adequately addressed all my concerns

Reviewer #2 (Remarks to the Author):

To authors,

This is a revised submission of an interesting study by Dr. Puri and the colleagues aiming at understanding the role of Sox9 in mature pancreatic beta cells. The authors had addressed most of the reviewer's concerns and provide new data to support the conclusions. However, there are still some points need to be addressed by the authors:

1. The authors still failed to provide data to elucidated mechanism that links SOX9 to disrupted alternative splicing patterns. As the authors explained low level of Sox9 protein in islets is the major issue, have the authors detected Sox9 protein level in beta cell lines such as Min6, β TC-6 etc.?
2. Related to question 1, as the authors mentioned in the manuscript, the Sox9 level is too low to be detectable by current technology and the authors believed that low levels of gene expression can also be sufficient to effect downstream mechanisms and influence cell function, and they listed Ngn3 as an example. I agree that low level of transcription factors can exert profound effect through regulating a large number of downstream targets genes. But here the authors believed that they found a novel role of Sox9 in alternative splicing, which is distinct from its canonical DNA binding function as a transcription factor, maybe through a direct binding to the spliceosome. Therefore, such low level of Sox9 in beta cells make this hypothesis less convincing.
3. The authors suggested that deletion of Sox9 impaired the insulin secretion in pancreatic islets. However, from Fig.1K and Fig2A, it seems that the insulin content is significantly decreased in absence of Sox9, which indicated that reduced insulin synthesis may be responsible for impaired insulin secretion. Therefore, the authors should also provide the electron micrographs (EM) to confirm the effect of Sox9 on insulin secretion.
4. The authors suggested the Srsf family as a major target of Sox9 in beta cells. Since Sox9 expression in beta cells is significantly decreased under HFD feeding, whether Srsf family in pancreatic islets is also alternatively spliced under metabolic stress?

Reviewer #3 (Remarks to the Author):

Given the revised manuscript and the authors' responses, I do think adding it would be helpful to add Reviewer 3, Fig. 1 to the supplementary file, and to add the explanation of equivalent classes. My previous comments have been addressed by this revised version.

REVIEWER COMMENTS

We appreciate the reviewers' comments, and are pleased to read that two out of the three reviewers found the changes to the revised manuscript useful and successfully addressing all their concerns and questions. We have spent the last several months addressing the comment from Reviewer 2 regarding the underlying mechanism by which Sox9 regulates alternative splicing in beta cells, and are including our findings in the revised manuscript. We also address the reviewer's additional comments in a point-by-point manner below. We hope that with the newly added data as well as our responses to the concerns, we will successfully address all the points raised by all three reviewers. As before, our responses are in blue.

Reviewer #2 (Remarks to the Author):

To authors,

This is a revised submission of an interesting study by Dr. Puri and the colleagues aiming at understanding the role of Sox9 in mature pancreatic beta cells. The authors had addressed most of the reviewer's concerns and provide new data to support the conclusions. However, there are still some points need to be addressed by the authors:

1. The authors still failed to provide data to elucidated mechanism that links SOX9 to disrupted alternative splicing patterns. As the authors explained low level of Sox9 protein in islets is the major issue, have the authors detected Sox9 protein level in beta cell lines such as Min6, β TC-6 etc.?

We have indeed detected Sox9 expression in cell lines, including the rat insulinoma cell line INS-1 (please see included western blot), albeit at low levels. If the reviewer believes this will add to our findings, we can add this western blot to the supplemental figure 1.

2. Related to question 1, as the authors mentioned in the manuscript, the Sox9 level is too low to be detectable by current technology and the authors believed that low levels of gene expression can also be sufficient to effect downstream mechanisms and influence cell function, and they listed Ngn3 as an example. I agree that low level of transcription factors can exert profound effect through regulating a large number of downstream targets genes. But here the authors believed that they found a novel role of Sox9 in alternative splicing, which is distinct from its canonical DNA binding function as a transcription factor, maybe through a direct binding to the spliceosome. Therefore, such low level of Sox9 in beta cells make this hypothesis less convincing.

The idea that splicing and transcription happen simultaneously is not a new one. In fact, transcription factors such as PGC1 are found to colocalize with splicing factors within the nucleus (Monsalve et al., 2000, *Molecular Cell*). ERG factors, a subfamily of ETS transcription factors, are also involved in alternative splicing (Saulnier et al., 2021, *Nucleic Acids Research*). Indeed, previous work has implicated Sox9 to be such a factor (Girardot et al., 2018, *Nuclear Acid Research*, Penrad-Mobayed et al., 2018, *Scientific Reports*).

In addition to what is already reported in literature, we found that in both mouse and human systems, deletion of SOX9 led to change in alternative splicing, further confirming that this function of Sox9 is conserved across species, underscoring its relevance and importance. To address the reviewer's comment and to deepen our understanding of the mechanism, we carried out a Proximity Ligation Assay (DUO92101-1KT, Sigma). This assay allows for identification of protein interaction within fixed cells at the single molecule level. We chose to examine SOX9 proximity to Y14, a protein that localizes to the splicing machinery. Y14 forms part of the Exon Junction Complex (EJC) within the spliceosome (Zhang et al., 2017, *Cell*). A positive signal using the PLA assay is indicative of close association between the proteins, as is shown in **Fig. 3K** in the revised manuscript. Such strong colocalization within the nucleus places SOX9 in close proximity to Y14 (<40nm), a part of the splicing machinery - further strengthening our conclusion that SOX9 is associated with and plays a role in alternative splicing in rodent and human cells.

3. The authors suggested that deletion of Sox9 impaired the insulin secretion in pancreatic islets. However, from Fig. 1K and Fig2A, it seems that the insulin content is significantly decreased in absence of Sox9, which indicated that reduced insulin synthesis may be responsible for impaired insulin secretion. Therefore, the authors should also provide the electron micrographs (EM) to confirm the effect of Sox9 on insulin secretion.

We agree with the reviewer that doing a more in depth EM analysis will shed light on the insulin granule biogenesis, which we expect to be perturbed based on the findings that pro-insulin levels appeared increased upon Sox9 deletion as seen by immunostaining. As we see clear changes in the alternative splicing of factors involved in the ER protein processing and degradation pathway, it is not surprising that insulin processing is affected. However, as the reviewer can appreciate, we have had to generate animals that are significantly aged, and currently we have used all our older cohorts for the revision experiments including doing GSIS assays from isolated islets. The time required to generate a colony of animals to carry out EM will be significant and will be part of our follow up study in the future.

4. The authors suggested the Srsf family as a major target of Sox9 in beta cells. Since Sox9 expression in beta cells is significantly decreased under HFD feeding, whether Srsf family in pancreatic islets is also alternatively spliced under metabolic stress?

There is a large body of work that demonstrates a change in alternative splicing within pancreatic islets upon stress – both in T1D and T2D, cells show dysregulated alternative splicing. Among many others, the Eizirik lab has clearly demonstrated the change in alternative splicing and the Srsf family of regulators in beta cells that has consequences for metabolic regulation and implications for disease (Alvelos MI et al., PMID: PMC6148369; Juan-Mateu J. et al., PMID: PMC5828453).

To address the reviewer's question, we stained mouse islets from a high fat diet fed cohort for Srsf5 and compared it to normal chow fed animals. As shown in the accompanying figure, we can see

clear nuclear accumulation of Srsf5 in islets of the chow-fed animals. However, the cytoplasmic staining in islets from high fed diet-fed animals appears significantly increased, suggesting that in these animals, Srsf5 localization and most likely function is perturbed. While these preliminary data are interesting, they would require significantly more analysis that is beyond the scope of this manuscript and will be addressed in follow up studies.

Reviewer #3 (Remarks to the Author):

Given the revised manuscript and the authors' responses, I do think adding it would be helpful to add Reviewer 3, Fig. 1 to the supplementary file, and to add the explanation of equivalent classes. My previous comments have been addressed by this revised version.

As requested by the reviewer, we have added the data that show the proportions of different cell populations in the revised Supplementary Figure 5. We have also added the explanation of the equivalence classes to the text.

REVIEWERS' COMMENTS

Reviewer #1 (Remarks to the Author):

The authors have addressed all concerns Reviewer 2 previously had and have included additional data in the work that further supports their conclusions.

I am not clear how the INS-1 western blot could be easily integrated into Supplemental Figure 1, but if the authors think it would add to their study then they could include it.